# High-throughput barcoding of nanoparticles identifies cationic, degradable lipid-like materials for mRNA delivery to the lungs in female preclinical models

Lulu Xue [1], Alex G. Hamilton [1], Gan Zhao [2], Zebin Xiao [2], Rakan El-Mayta [1,3], Xuexiang Han [1], Ningqiang Gong [1], Xinhong Xiong[4], Junchao Xu [1], Christian G. Figueroa-Espada [1], Sarah J. Shepherd[1], Alvin J. Mukalel[1], Mohamad-Gabriel Alameh [3,5], Jiaxi Cui [4], Karin Wang [6], Andrew E. Vaughan [2], Drew Weissman[3,5] & Michael J. Mitchell [1,5,7,8,9,10] ✉

Lipid nanoparticles for delivering mRNA therapeutics hold immense promise for the treatment of a wide range of lung-associated diseases. However, the lack of effective methodologies capable of identifying the pulmonary delivery profile of chemically distinct lipid libraries poses a significant obstacle to the advancement of mRNA therapeutics. Here we report the implementation of a barcoded high-throughput screening system as a means to identify the lung-targeting efficacy of cationic, degradable lipid-like materials. We combinatorially synthesize 180 cationic, degradable lipids which are initially screened in vitro. We then use barcoding technology to quantify how the selected 96 distinct lipid nanoparticles deliver DNA barcodes in vivo. The top-performing nanoparticle formulation delivering Cas9-based genetic editors exhibits therapeutic potential for antiangiogenic cancer therapy within a lung tumor model in female mice. These data demonstrate that employing high-throughput barcoding technology as a screening tool for identifying nano-particles with lung tropism holds potential for the development of next-generation extrahepatic delivery platforms.

Lipid nanoparticles (LNPs) are clinically relevant delivery agents capable of delivering messenger RNA (mRNA)-based therapeutics, holding great promise for use in vaccination[1,2], protein replacement therapy[3,4], cancer immunotherapy[5,6], and CRISPR-Cas-based gene editing[7,8].

Recently, the US Food and Drug Administration (FDA) fully approved two mRNA vaccines against COVID-19 enabled by LNPs[9]; clinical trials have also demonstrated robust in vivo CRISPR gene editing to treat hereditary transthyretin amyloidosis in patients[10], which represents a

[1]Department of Bioengineering, University of Pennsylvania, Philadelphia, PA 19104, USA. [2]Department of Biomedical Sciences, School of Veterinary Medicine, University of Pennsylvania, Philadelphia, PA 19104, USA. [3]Department of Medicine, University of Pennsylvania, Philadelphia, PA 19104, USA. [4]Yangtze Delta Region Institute (Huzhou), University of Electronic Science and Technology of China, Huzhou, Zhejiang 313001, China. [5]Penn Institute for RNA Innovation, Perelman School of Medicine, University of Pennsylvania, Philadelphia, PA 19104, USA. [6]Department of Bioengineering, Temple University, Philadelphia, PA 19122, USA. [7]Abramson Cancer Center, Perelman School of Medicine, University of Pennsylvania, Philadelphia, PA 19104, USA. [8]Institute for Immunology, Perelman School of Medicine, University of Pennsylvania, Philadelphia, PA 19104, USA. [9]Cardiovascular Institute, Perelman School of Medicine, University of Pennsylvania, Philadelphia, PA 19014, USA. [10]Institute for Regenerative Medicine, Perelman School of Medicine, University of Pennsylvania, Philadelphia, PA 19104, USA. ✉e-mail: mjmitch@seas.upenn.edu

significant milestone for mRNA therapeutics. Unfortunately, there have also been clinical failures driven in part by inefficient delivery[11,12]. These advancements and failures underscore the need to develop potent ionizable lipids to facilitate efficient mRNA delivery for disease treatments; however, extrahepatic mRNA delivery has remained challenging, owing to the slow blood flow and discontinuous vasculature of hepatic sinusoids that enhance liver delivery[13–16].

The lungs represent a compelling target for mRNA delivery due to the diverse range of pathological targets affecting endothelial[17,18], epithelial[19,20], and immune cells[21,22] in lung-associated diseases. Various approaches have been utilized to target the lungs and their respective cellular populations[13,14,23–29], including pre-treating animals to overwhelm the liver[30] or reduce drug activity[31] in specific cell types to shift tropism, conjugating receptor ligands onto LNPs surface for active targeting[32], and interacting with serum proteins for endogenous targeting[33,34]. Although some of these approaches have resulted in advanced phase 1/2 clinical studies[11,35], the multistep strategy and safety concerns have limited their applicability to evaluate large lipid libraries. To identify the lead performer ionizable lipids for mRNA delivery in each large, chemically distinct lipid library, scientists must explore the transfection efficacy of each LNP formulation in delivering its payload into target tissues and cells in vivo. Because injecting and sacrificing thousands of mice per lipid library is challenging, typically only a fraction of the LNP candidates can be evaluated in vivo, limiting the amount of data available for the remainder of the initial large lipid library. It has also been reported that in vitro delivery profiles are usually poor predictors of in vivo nanoparticle delivery[36]; thus, development of high-throughput methods for in vivo screening can accelerate the discovery of LNPs with chemical structure and properties that can overcome delivery barriers for gene therapy applications in the lungs[37].

Here, we employed a barcoded DNA (b-DNA)-based high-throughput LNP screening system[38,39], which allows the investigation of many nanoparticles in a single animal, to explore a combinatorial cationic degradable (CAD) lipid library assessing ionizable lipid chemical structure-activity relationships for pulmonary delivery (Fig. 1a). We initially formulated 180 CAD LNPs with mRNA encoding firefly luciferase (FLuc) to study their delivery potential using in vitro high-throughput screening. We then selected 96 CAD LNPs to encapsulate b-DNA and FLuc mRNA, pooled the LNP formulations, and systemically administered this pool into mice. We then extracted DNA to quantify accumulation in different organs (heart, liver, spleen, lungs, and kidneys) through deep sequencing, identifying 21 promising LNP candidates for pulmonary delivery. The top 4 LNPs formulated with FLuc mRNA were further counterscreened in mice to evaluate mRNA delivery efficacy in the lungs. We identified LNP-CAD9 as the top performing LNP to deliver FLuc mRNA to the lungs in vivo, demonstrating luciferase expression preferentially in the lungs (~90% of total luminescence flux). This LNP delivering Cre mRNA can preferentially edit lung endothelial cells at a dose of 0.3 mg kg$^{-1}$. Moreover, LNP-CAD9 co-delivering Cas9 mRNA/VEGFR2 single guide RNA (sgRNA) effectively induced VEGFR2 knock out in lung endothelial cells of female mice at a dose of 4.0 mg kg$^{-1}$ and thus demonstrated significant therapeutic potential in antiangiogenic therapy for suppressing tumor growth within a lung tumor model, outperforming a gold standard lung-tropic MC3/DOTAP LNP system. This proof-of-concept study suggests that high-throughput barcoding technology can be utilized as a screening tool for identifying structurally distinct nanoparticles for extrahepatic mRNA delivery to the lungs.

## Results

### Combinatorial design of CAD lipids

It has been reported that incorporating atypical chemical motifs can alter protein corona composition on LNPs and shift organ tropism[14,33,40]. Notably, positively charged molecules enable the binding of distinct proteins which can interact with specific cellular receptors highly expressed within the lungs for extrahepatic nucleic acid delivery. These molecules are typically utilized as a fifth constituent incorporated into the LNP formulation for tissue-specific mRNA delivery; however, identifying the interplay between the structure of the lipids themselves and lung tropism remains challenging.

We rationally designed ionizable lipids through "Schiff base reduction" that links amine heads and aldehyde degradable alkyl tails (Fig. 1b)[41]. In brief, combinatorial reactions between 12 amine heads and 15 aldehyde degradable tails were conducted for 3 h to yield Schiff base intermediates under acetic acid (AcOH) (Fig. 1c, Supplementary Figs. 1–18). A subsequent reduction under sodium borohydride (NaBH$_4$) was conducted for 1 h and led to the final cationic degradable (CAD) lipids. We broadened this library by varying amine core structures, tail architecture, tail substitution numbers, and tail lengths, giving the resulting 180 CAD lipids the nomenclature X-A$_y$-Z, where "X" indicates the order of amine cores in this study, "A$_y$-Z" represents aldehyde degradable tails ("y" represents the tail number; "Z" represents the carbon number on each tail). This "two-step, one-pot" reaction is simple and robust, yielding CAD lipids in several hours, which is significantly faster than the widely used Michael addition reaction[15,42]. Moreover, the final product can be used without further purification (Supplementary Fig. 18). We envision that this combinatorial CAD ionizable lipid library could extend the chemical diversity of ionizable lipid formulations for nucleic acid delivery applications.

### In vitro high-throughput screening to identify lipid-like materials for potent mRNA transfection

We initially investigated the structure-activity relationship (SAR) of CAD lipid-like materials for mRNA delivery in vitro. CAD LNPs encapsulating firefly luciferase (FLuc) mRNA were used to transfect HeLa cells. CAD LNPs were formulated using CAD lipids, the phospholipid DOPE, cholesterol, and lipid-anchored poly(ethylene glycol) (C14PEG2K) (35:16:46.5:2.5 molar ratio) and were mixed with FLuc mRNA via perfusion through a microfluidic mixing device designed with staggered herringbone features (Fig. 2a)[43,44]. The resulting CAD LNPs showed mRNA encapsulation efficiencies ranging from 74% to 95% (Supplementary Fig. 19). CAD LNPs showed uniform solid core morphology when investigated using cryo-transmission electron microscopy (cryo-TEM) (Fig. 2b, c). Additionally, all CAD LNPs exhibited low cytotoxicity (cell viability >85%) (Supplementary Fig. 20).

From in vitro screening in HeLa cells, we generated a heatmap of mRNA delivery by CAD lipids by calculating the relative hit rate (relative luminescence units, RLU >100) of different CAD lipid parameters to evaluate which structural parameters are most important for mRNA delivery in vitro (Fig. 2d). We initially investigated whether the cationic amine number of each CAD lipid influenced mRNA delivery efficacy, observing that CAD lipids with two secondary amines per lipid exhibited the highest mRNA delivery efficacy, with a hit rate of ~13% over the whole library (Fig. 2e). We postulated that CAD lipids with a greater number of secondary amine groups (>2) had a relatively higher binding affinity with mRNA, making it difficult to release hydrophilic mRNA compounds into the cytoplasm for efficient delivery. Additionally, we observed that CAD lipids with branched architecture exhibited substantially higher mRNA delivery than linear structured ones (Fig. 2f), which corroborates a previous report that branched tails may increase endosomal escape for mRNA delivery[45]. Importantly, tail number and tail length on each aldehyde were very influential for mRNA delivery, where a tail number of 2 and a tail length of 7 on each aldehyde resulted in the highest hit rates (Fig. 2g, h). These observations are in accordance with previously reported LNP systems, where efficacy generally correlated with tail substitution sites and diversity[46,47]. From this library, we then selected 96 CAD LNP formulations which showed effective transfection in HeLa cells for subsequent in vivo studies.

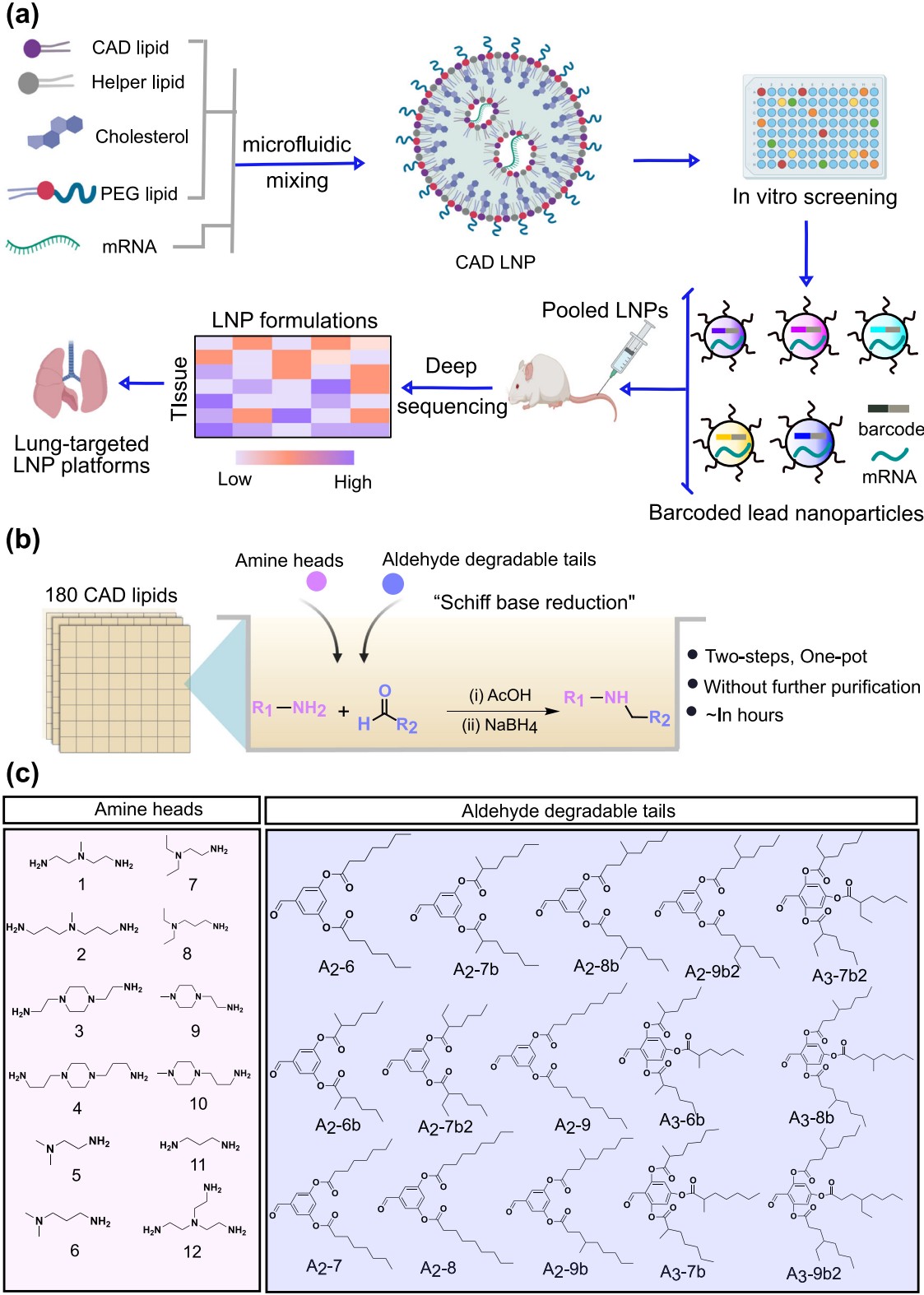

**Fig. 1 | High-throughput LNP screening facilitates the discovery of cationic degradable (CAD) lipid-like materials for mRNA delivery to the lungs. a** CAD LNPs were initially formulated using a microfluidic mixing device by mixing nucleic acid with CAD lipids, helper lipid, cholesterol, and PEG-lipid. Following in vitro high-throughput screening, a series of CAD LNPs were selected and formulated to co-encapsulate b-DNA and mRNA, pooled, and systemically administered into mice, allowing for quantification of accumulation in each organ (heart, liver, spleen, lungs, and kidneys) using deep sequencing to identify CAD lipid candidates for lung-targeted mRNA delivery. **b** A combinatorial library of CAD lipids was chemically synthesized through "Schiff base reduction" by reacting amine heads and aldehyde degradable tails. **c** Overview of 12 amines cores and 15 aldehyde degradable tails used to synthesize 180 CAD lipids. **a** Created with BioRender.com.

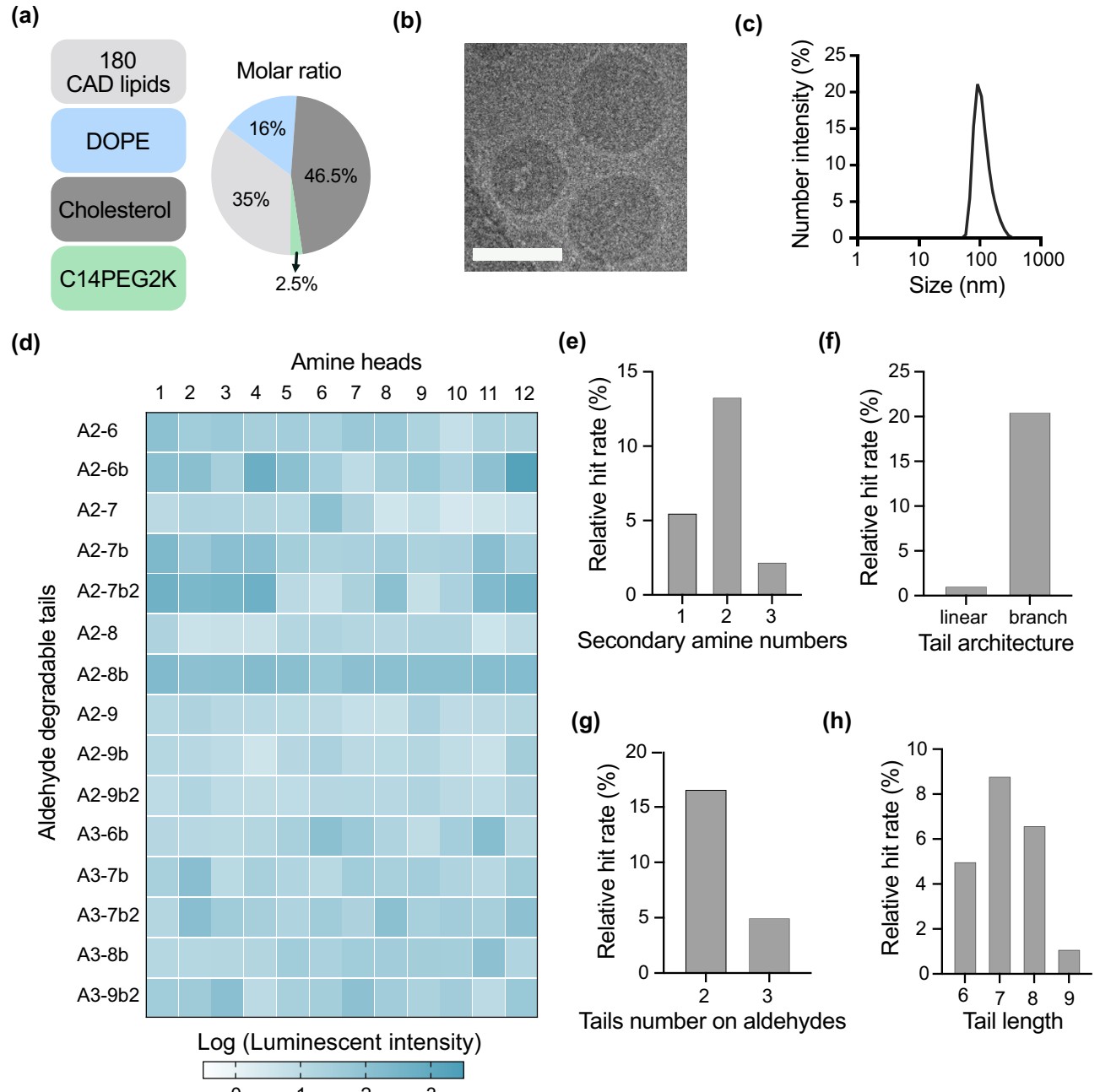

**Fig. 2 | Investigation of structure-activity relationship of CAD LNPs for firefly luciferase (FLuc) mRNA delivery in vitro. a** CAD LNP formulation parameters. CAD LNPs were formulated with one of 180 distinct CAD lipids, DOPE, cholesterol, and C14PEG2K at a molar ratio of 35:16:46.5:2.5, for a total of 180 distinct LNP formulations. **b** Representative cryogenic transmission electron microscopy (cryo-TEM) image of 3-A$_2$−7b LNP morphology ($n$ = 3 replicates). Scale bar: 100 nm. **c** Hydrodynamic size distribution of 3-A$_2$-7b LNP obtained by dynamic light scattering (DLS). **d** A heatmap of luciferase expression following treatment of HeLa cells with CAD LNPs (10 ng luciferase mRNA, $n \geq 3$ replicates). Relative luminescence unit (RLU) values of > 100 were classified as hits for hit rate calculation. **e** Relative hit rate of CAD LNPs with different secondary amine numbers. **f** Relative hit rate of CAD LNPs with different tail architectures. **g** Relative hit rate of CAD LNPs with different tail substitution numbers on each aldehyde. **h** Relative hit rate of CAD LNPs with different tail lengths. Source data are provided in the Source Data file.

## Understanding CAD lipid structure and organ tropism relationships in vivo

To better understand the relationship between CAD lipid structure and their organ tropism, we evaluated 96 CAD LNPs in vivo through a b-DNA-based assay, which can quantify how hundreds of different LNPs deliver mRNA in vivo[38,39]. LNP-CAD1, with chemical composition 1, was formulated to carry b-DNA 1 and FLuc mRNA, and LNP-CADN, with chemical composition N, to carry b-DNA N and FLuc mRNA, at a weight ratio of 10:1 (Fig. 3a). To investigate the potential influence of b-DNA on LNP structure and the efficacy of mRNA delivery in vitro, the 3-A$_2$-7b

LNP was used as a representative example to encapsulate b-DNA/FLuc mRNA (at a weight ratio of 10:1) and FLuc mRNA, respectively. These two LNPs exhibit similar structural morphology (Fig. 2b and Supplementary Fig. 21a); however, LNP carrying b-DNA/FLuc mRNA displayed a reduction in average particle size (Fig. 2c and Supplementary Fig. 21b), which is likely attributable to the smaller size of b-DNA compared to mRNA. Nonetheless, despite these differences in particle size, no appreciable distinction in mRNA transfection efficacy was observed between these two LNP formulations in vitro (Supplementary Fig. 21c).

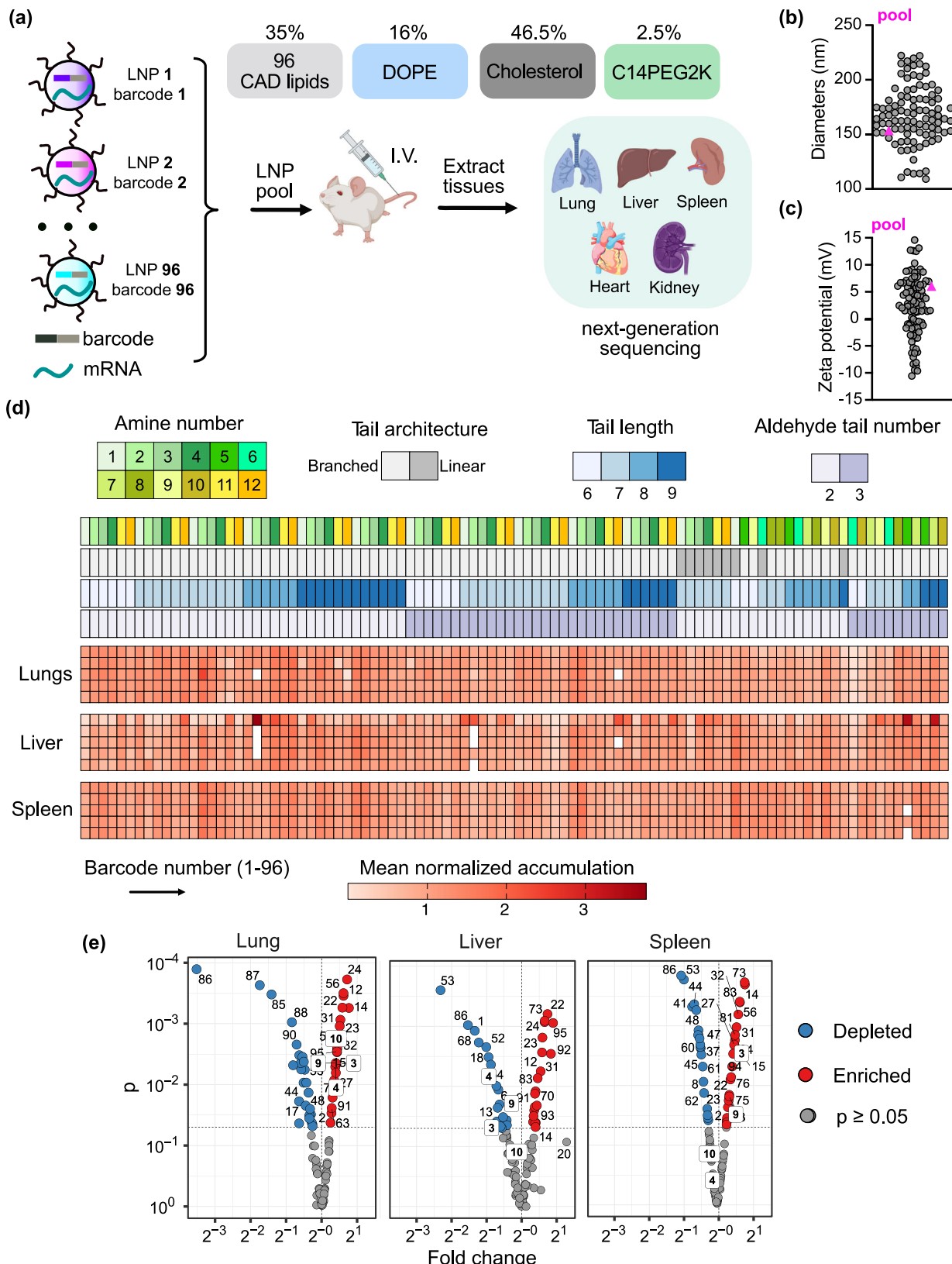

By integrating unique DNA barcodes in each LNP, we were able to assess the organ tropism of each tested LNP through deep sequencing. The hydrodynamic diameter of these LNPs was evaluated as a quality control measure, showing that the size of these LNPs ranged from 100 to 250 nm (Fig. 3b and Supplementary Tables 1, 2), consistent with previous reports that pipette-mixed LNPs are generally larger in size[38,48].

As expected due to the use of CAD lipids, over 65% of resulting CAD LNPs exhibited a positive charge (Fig. 3c and Supplementary Tables 1, 2). Furthermore, we tested the diameter and zeta potential of the pool of CAD LNPs and found them within the range of the diameter and zeta potential of the 96 individual CAD LNPs, respectively, indicating that mixing the CAD LNPs did not adversely affect solubility (Fig. 3b, c).

**Fig. 3 | In vivo structure-activity relationship analysis of 96 chemically distinct CAD lipids and organ tropism. a** Schematic illustration of barcoding approach to probe biodistribution profile of designed CAD LNPs. LNPs were formulated by pipette mixing to encapsulate barcoded DNA (b-DNA) and FLuc mRNA. Lipid phases consisted of one of 96 CAD lipids, DOPE, cholesterol, and C14PEG2K at a molar ratio of 35:16:46.5:2.5. LNP formulations were pooled together and administered systemically to C57BL/6 J female mice ($n = 5$). Tissues were isolated 6 h post-administration, DNA was extracted, and accumulation of b-DNAs was quantified by deep sequencing. I.V.: intravenous. **b** Hydrodynamic diameter of all administered CAD LNPs. The diameter of the LNP pool control (pink triangle symbol) falls within the range of the CAD LNPs composing the pool. **c** Zeta potential of all administered CAD LNPs. The zeta potential of the pooled LNP (pink triangle symbol) falls within the range of the CAD LNPs composing the pool. **d** Heatmap visualizing accumulation of CAD LNPs in different organs as measured by deep sequencing. Dark clusters represent higher relative accumulation of b-DNA in a specific tissue sample. Structural details of CAD lipids used in each LNP formulation are described above the heatmap. **e** Volcano plots summarizing enrichment analysis of barcodes in the lungs, liver, and spleen. The normalized accumulation of LNP formulation was compared to the aggregate LNP pool (i.e., basemean) using a two-sided Wilcoxon rank-sum test. False discovery rate was controlled using the Benjamini-Hochberg method. The exact $P$ values from each comparison are provided in Supplementary Data 1. **a** Created with BioRender.com. Source data are provided in the Source Data file.

After characterizing the pool of 96 CAD LNPs, we then intravenously (i.v.) administered them in C57BL/6 J female mice at a total nucleic acid dose of 1.0 mg kg$^{-1}$ (averaging 0.01 mg total nucleic acid/kg/particle, for all 96 CAD LNPs), isolated tissues (heart, liver, spleen, lung, and kidney) 6 h post-injection and extracted DNA from these tissues (Fig. 3a and Supplementary Table 3). Extracted DNA samples were amplified by polymerase chain reaction (PCR) and deep sequenced to compare the relative accumulation of CAD LNPs in different tissues through comparison to the uninjected LNP pool. This approach allowed us to identify CAD LNPs with preferential accumulation in specific organs.

We then used this large dataset to analyze a comprehensive in vivo structure-activity relationship. A heatmap was generated based on the normalized accumulation of each barcoded oligomer in different organs (Fig. 3d and Supplementary Fig. 22). Within the heat map, darker red represents greater relative accumulation in a tissue of interest. The secondary amine number of each CAD lipid, tail architecture, tail length, and tail number on each aldehyde group appeared to significantly affect lipid activity. Specifically, more secondary amine-based CAD lipids (secondary amine number ≥2) preferentially delivered cargoes to the lungs compared to monoamine CAD lipids (Fig. 3d). We hypothesized that more strongly cationic CAD lipids may result in a relatively positive charge in the LNP formulation, which is supported by the fact that ~65% of these LNPs display a positive zeta potential (Supplementary Tables 1–2). This conclusion is also in accordance with previous studies reporting that cationic lipids facilitate LNP formulations to deliver genetic cargo into the lungs[14,33,40]. We further investigated the correlation between LNP size and lung delivery efficacy, observing only a weak relationship between LNP size and lung-tropic activity (Supplementary Fig. 23). The lung tropism of these LNPs may derive from alternative mechanisms such as endogenous targeting[14,33]. To visualize the in vivo activity of tested CAD LNPs in more detail, we generated volcano plots to show the results of enrichment analysis (Fig. 3e)[49]. Lung-targeted delivery carriers demonstrated a pronounced preference for lung distribution, while their presence in the liver and spleen was either significantly reduced or not notably enriched. Through this enrichment analysis, we found that 21 of our tested CAD LNPs can efficiently deliver nucleic acid cargo into the lungs. Notably, LNP-CAD24 and LNP-CAD56 exhibited the greatest enrichment within the lungs. However, both LNP-CAD24 and LNP-CAD56 displayed substantial enrichment in the liver and spleen, which is not desired for lung-targeted delivery. Consequently, LNP-CAD3, LNP-CAD4, LNP-CAD9, and LNP-CAD10 emerged as primary candidates selected from the b-DNA screening for lung-specific delivery, as they demonstrated highly enriched accumulation in the lungs with depleted or not notably enriched delivery in the liver and spleen (Fig. 3e), bringing these LNP formulations to the fore in our search for LNPs for mRNA delivery to the lungs.

## Validation of top performing LNPs for mRNA delivery to the lungs

Through high-throughput screening both in vitro and in vivo, we screened 180 chemically distinct CAD LNPs to discover that 21 of them showed in vivo delivery to the lungs. By further enrichment analysis, we identified 4 lead LNP formulations to efficiently deliver nucleic acid cargo to the lungs (Fig. 4a). To verify that measures of b-DNA accumulation can accurately identify CAD LNPs for mRNA delivery, we used the lead identified liver formulation, LNP-CAD20 (Fig. 3d, e), to deliver FLuc mRNA in vivo. The specific luciferase expression in the liver supported our high-throughput barcoded screening results (Supplementary Fig. 24). To further validate mRNA delivery efficacy to the lungs by the lead lung-tropic LNP formulations, LNP-CAD3, 4, 9, and 10 were formulated with FLuc mRNA and systemically injected into C57BL/6J female mice at a dose of 0.1 mg kg$^{-1}$ (Fig. 4b). Bioluminescence imaging confirmed that the selected 4 CAD LNPs can functionally deliver mRNA the lungs as expected from high-throughput screening results. By analyzing luminescence of the lungs, liver, and spleen, we identified LNP-CAD9 as the top performing LNP candidate for pulmonary mRNA delivery (Fig. 4c–f and Supplementary Tables 1, 2), with luciferase expression predominantly in the lungs (~90% of total luminescence flux).

Next, we explored whether LNP-CAD9 delivered mRNA at a clinically relevant dose[50]. For this, we employed genetically engineered tdTomato reporter mice, an Ai14 (constitutive loxP-STOP-loxP-tdTomato) mouse model[51], which have gained widespread use in organ-specific gene editing applications[52,53]. These mice feature a loxP flanked stop cassette that effectively prevents expression of tdTomato protein[51–53]. LNPs delivering Cre mRNA into specific organs have the capability to delete the stop cassette, thereby producing tdTomato fluorescence only in transfected cells upon intracellular delivery of Cre recombinase mRNA[28,40,54] (Fig. 4g). Following a single administration of 0.3 mg kg$^{-1}$ Cre mRNA using LNP-CAD9, efficient lung gene editing was observed 3 days post-administration (Fig. 4h and Supplementary Fig. 25), resulting in ~60% tdTomato$^+$ endothelial cells. To benchmark the lung delivery potency of the identified LNP-CAD9, a gold standard lung-tropic MC3/DOTAP LNP system[23,33,40], in which DOTAP was reported to be incorporated as a cationic lipid component for facilitating mRNA delivery to the lungs by the FDA-approved MC3 LNP, was produced and tested. LNP-CAD9 induced a significantly higher percentage of tdTomato$^+$ endothelial cells (Fig. 4h and Supplementary Fig. 25), with a 2.7-fold increase compared to the MC3/DOTAP LNP, which was further validated by increased tdTomato area from immunostaining (Fig. 4i). LNP-CAD9 carrying Cre mRNA mainly reached the capillary endothelial cells of the vasculature in the lungs (Fig. 4i). Moreover, LNP-CAD9 delivering Cre mRNA did not demonstrate discernible editing of endothelial cells within the liver and heart, highlighting the specific and targeted editing of lung endothelium achieved by LNP-CAD9 (Supplementary Figs. 26, 27). Moreover, LNP-CAD9 displayed minimal in vivo toxicity in tissue section histology (Supplementary Fig. 28), demonstrating the promising translational potential of CAD LNPs. These results led us to conclude that LNP-CAD9, discovered by high-throughput screening technology, preferentially delivered mRNA into lung endothelial cells, substantially outperforming a gold-standard lung-tropic MC3/DOTAP LNP formulation.

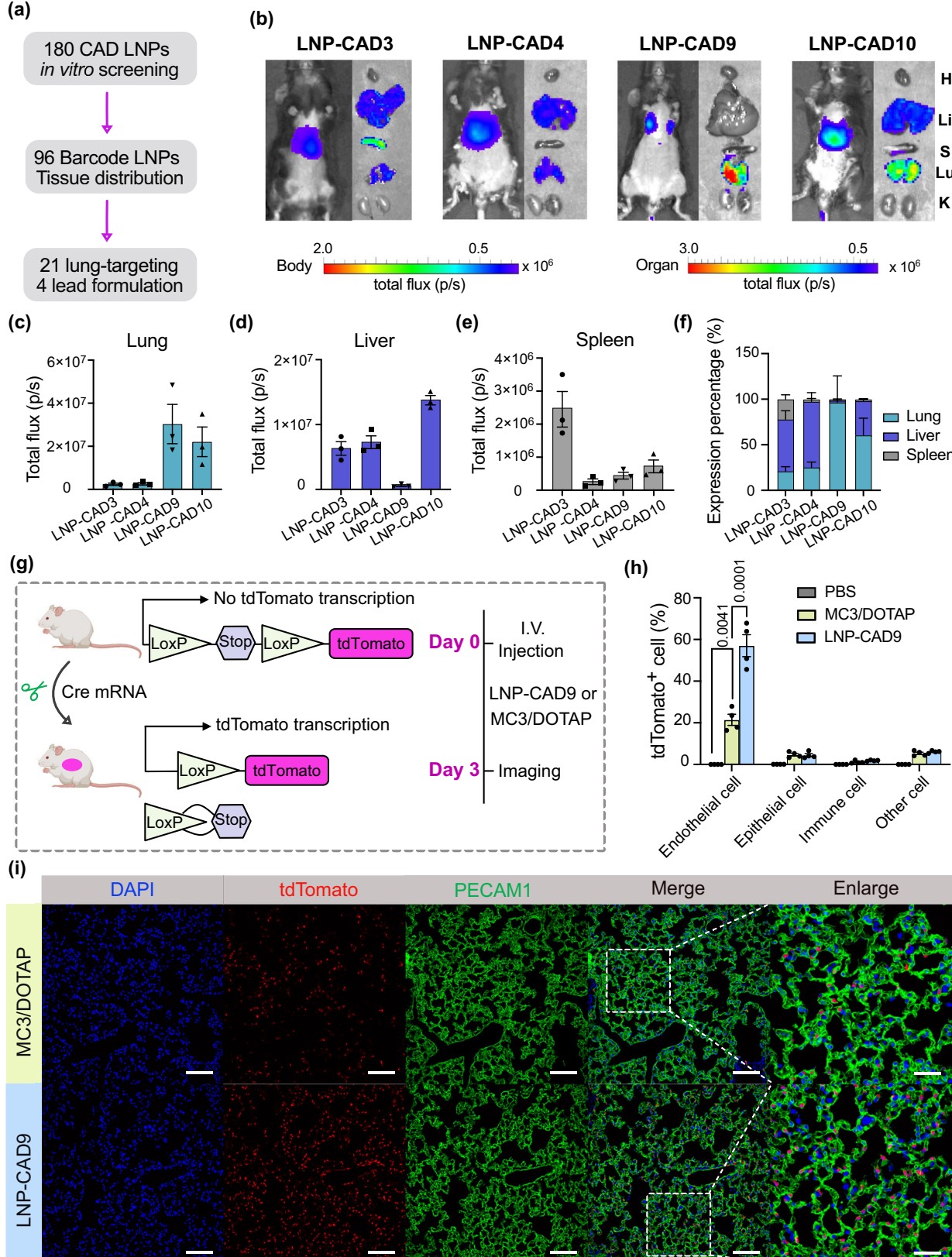

## Exploring therapeutic potential of top performing LNPs for in vivo gene editing for antitumor therapy

As LNP-CAD9 demonstrated promising editing efficacy of lung endothelial cells in vivo, we assessed the therapeutic potential of this platform in vascular-relevant disease models. Angiogenesis is a complex and vital physiological process, playing crucial roles in embryo development, wound healing, and collateral vessel formation[55,56]. However, angiogenesis becomes aberrantly upregulated in tumorigenesis, supporting tumor progression by supplying essential oxygen and nutrients[56,57]. Consequently, antiangiogenic therapy strategies aim to starve tumors by disrupting the delivery of these vital resources, thereby suppressing tumor growth, invasion, and metastasis[56,57].

**Fig. 4 | Validation of lead LNP formulations for mRNA delivery to the lungs of female mice. a** Four lead LNP formulations were discovered from a 180-CAD lipid library after high-throughput in vitro and in vivo screening. **b** Whole body and ex vivo imaging of luciferase expression mediated by LNP-CAD3, 4, 9, and 10 at 6 h post-injection (0.1 mg kg$^{-1}$ FLuc mRNA, $n = 3$ mice). H: heart, Li: liver, S: spleen, Lu: lung, K: kidney. Quantification of luciferase expression in the lungs (**c**), liver (**d**), and spleen (**e**) using region-of-interest (ROI) analysis. **f** Relative luciferase expression in each measured organ. **g** Ai14 mice were treated with LNP-CAD9 or MC3/DOTAP LNP encapsulating Cre mRNA 3 days prior to analysis (0.3 mg kg$^{-1}$, $n = 4$ mice). Lungs were digested and stained for quantifying cell populations for tdTomato

expression. PBS was injected as negative control. I.V.: intravenous. **h** Proportion of tdTomato$^+$ cells in the lung assessed by flow cytometry. **i** Representative immunostaining demonstrating signal overlap between tdTomato$^+$ cells and the endothelial cell marker platelet endothelial cell adhesion molecule 1 (PECAM1). DAPI was used for nuclear staining. Scale bars: 100 μm for the lung section images and 30 μm for enlarged images. **g** Created with BioRender.com. Statistical significance in (**h**) was calculated using one-way analysis of variance (ANOVA), followed by Dunnett's multiple comparison test. **$P < 0.01$; ***$P < 0.001$. Data are presented as mean ± s.e.m. Source data are provided in the Source Data file.

Among the numerous pro-angiogenic factors that have been discovered, vascular endothelial growth factor (VEGF) ranks as one of the most important[58]. VEGF binds to its receptor VEGFR2 on tumor vascular endothelial cells, which subsequently promotes endothelial cell proliferation, migration and survival, leading to increased tumor vascularization through the growth of new blood vessels[58]. Thus, the targeted disruption of VEGFR2 expression holds promise as an approach to inhibit the VEGF-VEGFR2 signaling pathway for antiangiogenic cancer therapy.

We established a representative orthotropic lung cancer model in female mice by i.v. administering Lewis lung carcinoma cell lines expressing GFP (LLC-GFP) and evaluated in vivo antiangiogenic cancer therapy efficacy of LNP-CAD9 co-encapsulating Cas9 mRNA/VEGFR2 single guide RNA (sgRNA) (Fig. 5a). MC3/DOTAP LNPs served as a benchmark lung-tropic LNP control. After tumor inoculation for 20 days, the mice were randomly allocated into four groups and received i.v. administration of PBS (G1), LNP-CAD9 co-delivering Cas9 mRNA/scrambled sgRNA (G2), LNP-CAD9 co-delivering Cas9 mRNA/VEGFR2 sgRNA (G3), or MC3/DOTAP co-delivering Cas9 mRNA/VEGFR2 sgRNA (G4), with a total RNA dosage of 4.0 mg kg$^{-1}$ (administered in two days of 2.0 mg kg$^{-1}$ each injection). Cas9/VEGFR2 sgRNA complex would be expected to induce double-strand breaks and insertions/deletions within the VEGFR2 locus and thereby inhibit the VEGF-VEGFR2 signaling pathway[59,60] (Fig. 5b). Seven days after the last administration, mice were euthanized, and their lungs were isolated to assess in vivo antitumor efficacy under the various treatment conditions. The mice that received LNPs co-delivering Cas9 mRNA/VEGFR2 sgRNA (G3 and G4) exhibited decreased VEGFR2 expression as quantified by RT-qPCR compared to PBS or scrambled sgRNA treated groups (G1 and G2) (Fig. 5c). Furthermore, evaluations of the tumor area per lung and histological examinations via hematoxylin and eosin (H&E) staining showed that the administration of LNP-CAD9 co-delivering Cas9 mRNA/VEGFR2 sgRNA led to a significant reduction in lung tumor burden, outperforming MC3/DOTAP LNP treated groups (Fig. 5d, e). The survival analysis also demonstrated that administration of LNP-CAD9 co-delivering Cas9 mRNA/VEGFR2 sgRNA presented the highest tumor-inhibitory potential among all treatments. This treatment regimen extended the median survival period from 32 days (G1) to 52 days (G3) (Fig. 5f). Additionally, we assessed angiogenesis of tumor tissues by immunostaining vascular endothelial cells using CD31 antibody[61] (Fig. 5g, h). The results showed a marked reduction in newly formed tumor blood vessels within the tumor site after treatment with LNP-CAD9 co-delivering Cas9 mRNA/VEGFR2 sgRNA (Fig. 5g). Quantification of microvascular density (MVD) in the tumor tissues showed that the administration of Cas9 mRNA/VEGFR2 sgRNA encapsulated by LNPs (G3 and G4) significantly reduced MVD compared to PBS or scrambled sgRNA treated groups (G1 and G2), indicating a pronounced antiangiogenic effect (Fig. 5h). Importantly, mice treated with LNP-CAD9 co-delivering Cas9 mRNA/VEGFR2 sgRNA demonstrated superior antitumor efficacy compared to MC3/DOTAP LNPs. Collectively, these findings underscore the therapeutic potential of LNP-CAD9 platform for inhibiting tumor angiogenesis in the lung, resulting in effective suppression of tumor growth and surpassing the lung-tropic gold-standard MC3/DOTAP LNP formulation.

## Discussion

Although mRNA therapeutics have made significant progress through systemic and local administration in preclinical/clinical trials[16,62,63], extrahepatic mRNA delivery still poses a major challenge[31,33,40]. The development of ionizable lipids has been identified as means to facilitate efficient mRNA delivery for hepatic and extrahepatic applications[64]. However, identification of top performers from a chemically distinct lipid library for mRNA delivery is tedious, typically requiring multiple screening steps both in vitro and in vivo. Moreover, in vitro delivery profiles are poor predictors of in vivo nanoparticle delivery, and false negatives abound, especially for extrahepatic organs. Thus, the expeditious deployment of high-throughput screening technology to enhance screening efficacy for extrahepatic mRNA delivery is of paramount importance.

Due to the importance of chemical characteristics of LNPs in determining delivery behavior and the tenuous relationship between in vitro and in vivo performance[39,54,65], there is a need for simple and rapid high-throughput screening techniques. High-throughput barcoding technology has been used for quantifying how hundreds of different LNPs deliver mRNA in vivo[38,39]. Historically, this barcoding approach is usually based on exploring formulation parameters orthogonally with a small chemically distinct lipid library[38,39,54,65], leaving the inherent structure-activity relationship of ionizable lipids somewhat overlooked for extrahepatic mRNA delivery. We employed this barcoded high-throughput screening system to identify a CAD LNP capable of delivering mRNA to the lungs from a chemically diverse library of 180 CAD lipids. Evaluating the distribution of b-DNA in various organs (heart, liver, spleen, kidneys, and lungs) through deep sequencing is a common and essential step in the identification of potential lipid-like materials for nucleic acid delivery. Both our work and previous studies have demonstrated the potential of incorporating barcoded technology to accelerate mRNA delivery for various applications[38,39,54,65,66]. In a prior study, we employed ionizable lipids to encapsulate barcoded mRNA for in vivo delivery screening[66], with a focus on a well-characterized C12-200 LNP with various formulation parameters, rather than a chemically distinct lipid library. Furthermore, previous studies did not establish a correlation between the properties of these ionizable lipids and their organ-specific delivery[38,39,54,65,66]. The current study represents a comprehensive effort to synthesize a chemically distinct cationic lipid library in an efficient and expeditious manner, establishing a relationship between the intrinsic properties of chemically distinct CAD lipids and their lung-specific mRNA delivery capabilities, and elucidating the therapeutic potential of the LNP-CAD9 platform in a disease model.

Genetically engineered tdTomato reporter mice, such as Ai14 mice, have gained widespread utility in organ-specific gene editing applications[52,53]. LNPs delivering Cre mRNA into specific organs have the capability to excise the stop cassette, thereby enabling the activation of constitutive tdTomato expression, which allows for detection of gene edited cells[28,40,54]. In this study, LNP-CAD9 delivering Cre mRNA were shown to efficiently edit lung endothelial cells at a dose of 0.3 mg kg$^{-1}$. Based on the endothelial cell tropism of LNP-CAD9, we further investigated the therapeutic potential of this platform in

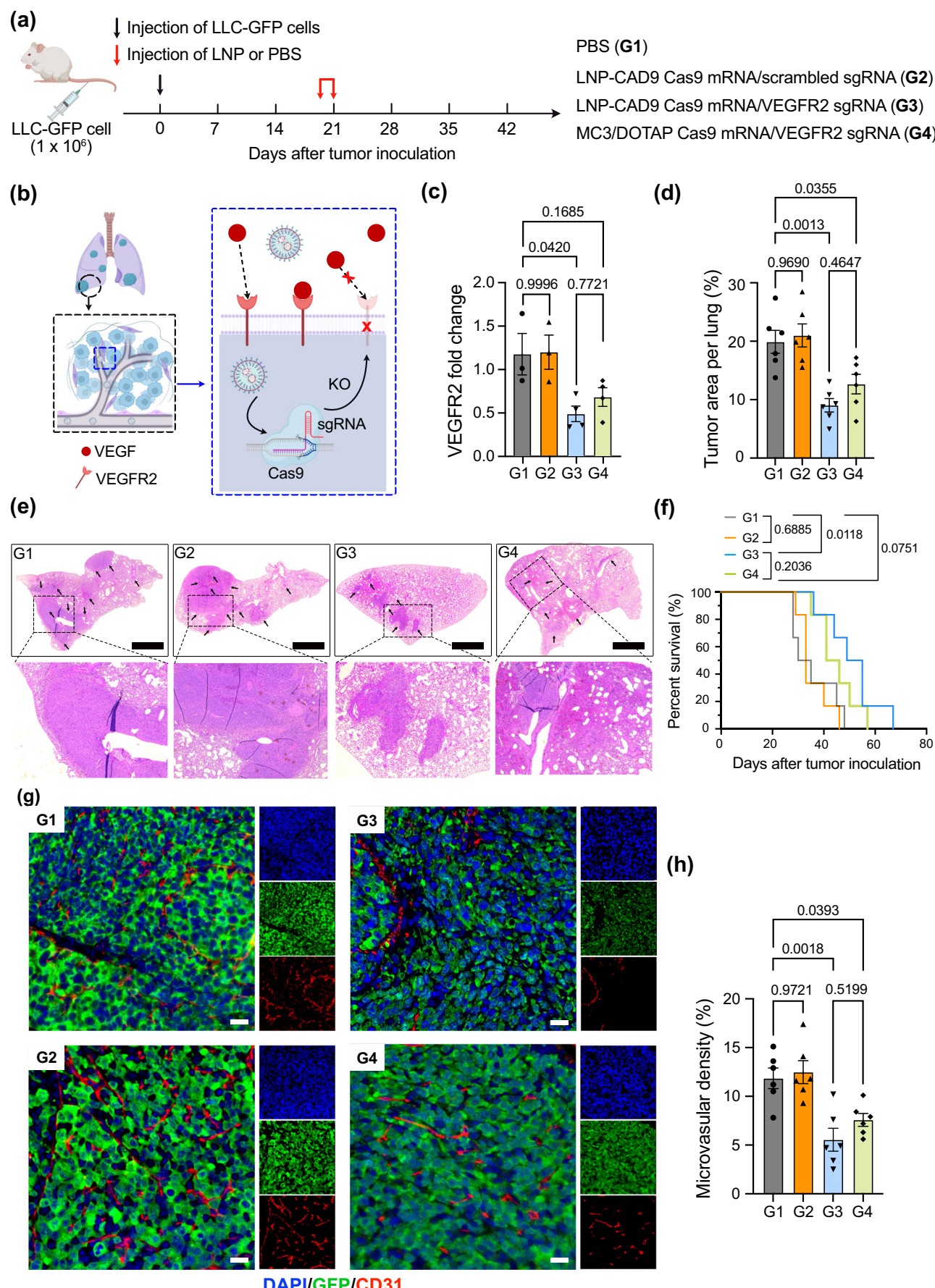

**Fig. 5 | Antiangiogenic therapy through knockout of VEGFR2 by LNPs in orthotropic lung cancer model. a** Schematic illustration of lung tumor implantation through i.v. injection of Lewis lung carcinoma cell lines expressing GFP (LLC-GFP) and treatment protocol of C57BL/6J female mice. On day 20 after tumor cell inoculation, mice were randomly assigned to four groups: PBS treated control (G1), LNP-CAD9 encapsulating Cas9 mRNA/scrambled sgRNA treatment (G2), LNP-CAD9 encapsulating Cas9 mRNA/VEGFR2 sgRNA treatment (G3), and MC3/DOTAP LNP encapsulating Cas9 mRNA/VEGFR2 sgRNA treatment (G4). Mice were treated every other day for a total of 2 doses (2.0 mg kg$^{-1}$ of RNA per injection). Seven days after the last administration, 6 mice in each group were euthanized and their lungs were isolated for antiangiogenic analysis. The remaining mice were used for survival evaluation. **b** Antiangiogenic mechanism through LNP-mediated VEGFR2 knockout. LNPs encapsulating Cas9 mRNA/VEGFR2 sgRNA demonstrated the ability to reduce the expression of VEGFR2 in lung endothelial cells, which inhibited the VEGF- VEGFR2 pathway. KO: knockout. **c** RT-qPCR measurement of VEGFR2 level in different treatment groups ($n = 3$ mice for G1 and G2 groups; $n = 4$ mice for G3 and G4 groups). **d** Quantification of tumor areas per lung of different treatment groups ($n = 6$ mice). **e** Representative H&E staining of lung tissue after sacrifice ($n = 6$ mice). Arrows indicate tumor areas in the lungs. Scale bar: 1 mm. **f** Percent survival of mice under different treatments ($n = 6$ mice). **g** Representative immunostaining of tumor areas in the lungs. Endothelial cells were stained by CD31 antibody. DAPI was used for nuclear staining. Scale bar: 100 μm. **h** Microvascular density (MVD) in the tumor area under different treatments ($n = 6$ mice). **a, b** Created with BioRender.com. Statistical significance in (**c**), (**d**), and (**h**) was calculated using one-way analysis of variance (ANOVA), followed by Dunnett's multiple comparison test. Statistical significance in (**f**) was calculated using a log-rank test. ***$p < 0.001$; **$p < 0.01$; *$p < 0.05$; $p > 0.05$, not significant. Data are presented as mean ± s.e.m. Source data are provided in the Source Data file.

an orthotopic lung cancer model, through CRISPR-Cas9 gene editor-mediated in vivo gene editing of VEGFR2 in lung endothelial cells for antiangiogenic cancer therapy. LNP-CAD9 co-delivering Cas9 mRNA and VEGFR2 sgRNA demonstrated remarkable in vivo antitumor therapy, as evidenced by a significant reduction in VEGFR2 levels, decreased tumor area per lung, prolonged survival, and a marked decrease in microvascular density within the tumor area, outperforming gold-standard MC3/DOTAP LNPs.

It has been reported that the apparent pKa of LNPs can exert influence over tissue-specific mRNA delivery activity. To investigate this, we selected representative liver-enriched (LNP-CAD20 and LNP-CAD95), lung-enriched (LNP-CAD10 and LNP-CAD3), and spleen-enriched (LNP-CAD14 and LNP-CAD73) LNPs for examination. In our investigation, LNP-CAD20 and LNP-CAD95 exhibited apparent pKa comfortably residing within the well-established range of 6 to 7 for liver-targeted LNPs, whereas LNP-CAD14 and LNP-CAD73 displayed lower pKa between ranging from 4 to 5 for spleen-targeted LNPs (Supplementary Fig. 29). These finding align with prior research investigating the relative pKa of LNPs concerning tissue-specific activity[33,62]. Nevertheless, in the case of lung-enriched LNPs (LNP-CAD10 and LNP-CAD3), our measurements revealed pKa of 4.905 and 5.806 (Supplementary Fig. 29), respectively. Notably, these values do not exceed the pivotal threshold of 9, as stipulated for effective lung-targeting LNP delivery in previous reports[33]. Cumulatively, these results provide compelling evidence that the pKa of LNPs represents only one facet of the complicated landscape governing tissue-specific mRNA delivery activity.

In summary, we designed a class of CAD lipids through a facile "Schiff base reduction" methodology, leading to synthesis of a combinatorial library of 180 chemically distinct CAD lipids in hours. We formulated these CAD lipids into CAD LNPs carrying FLuc mRNA to evaluate their transfection potential in HeLa cells in vitro. From these preliminary results, we identified 96 promising CAD lipids and used them to generate LNPs co-encapsulating b-DNA and FLuc mRNA, injecting these LNPs as a pool into female C57BL/6 J mice to enable quantification of cargo accumulation via deep sequencing. Enrichment analysis of the large resultant dataset identified 21 CAD LNPs with lung affinity, among which 4 LNPs demonstrated highly enriched pulmonary accumulation without enrichment in the liver and spleen, suggestive of strongly preferential pulmonary delivery. We employed low-throughput counterscreening to verify functional delivery by delivering mRNA encoding FLuc, identifying LNP-CAD9 as the lead candidate for pulmonary mRNA delivery, with ~90% of total luciferase expression observed in the lungs. Further investigation showed that LNP-CAD9 delivering Cre mRNA can preferentially edit lung endothelial cells at a dose of 0.3 mg kg$^{-1}$. Furthermore, this platform co-delivering Cas9 mRNA/VEGFR2 sgRNA effectively reduced VEGFR2 expression in lung endothelial cells, demonstrating its therapeutic potential in antiangiogenic therapy for suppressing tumor growth and prolonging mice survival within a lung tumor model of female mice, substantially outperforming a gold standard lung-tropic MC3/DOTAP LNP system. These findings demonstrate that high-throughput barcoding technology can be utilized as an efficient and effective screening tool for identifying structurally distinct nanoparticles for extrahepatic delivery to the lungs. We envision its potential application to enable mRNA delivery of chemically distinct lipid libraries for extrahepatic protein replacement, vaccine, and gene editing applications.

## Methods

### Materials
2,2′-Diamino-N-methyldiethylamine (98%, TCI America), 3,3′-diamino-N-methyldipropylamine (98%, TCI America), 2,2′-(piperazine-1,4-diyl)diethanamine (AmBeed), 1,4-bis(3-aminopropyl)piperazine (98%, TCI America), N,N-dimethylethylenediamine (98%, TCI America), N,N-dimethyl-1,3-propanediamine (99%, TCI America), N,N-diethylethylenediamine (98%, TCI America), N,N-diethyl-1,3-diaminopropane (99%, TCI America), 1-(2-aminoethyl)-4-methylpiperazine (97%, Alfa Aesar), 1-(3-aminopropyl)-4-methylpiperazine (98%, Alfa Aesar), 1,3-diaminopropane (99%, Sigma-Aldrich), tris(2-aminoethyl)amine (98%, TCI America), 3,5-dihydroxybenzaldehyde (AmBeed), 2,4,6-trihydroxybenzaldehyde (AmBeed), heptanoic acid (98%, TCI America), 2-methylhexanoic acid (98%, TCI America), n-octanoic acid (98%, TCI America), 2-methylheptanoic acid (98%, TCI America), 2-ethylhexanoic acid (99%, TCI America), nonanoic acid (98%, TCI America), 4-methyl-n-octanoic acid (98%, TCI America), decanoic acid (98%, TCI America), 4-methylnonanoic acid (98%, TCI America), 4-ethyloctanoic acid (98%, TCI America), 4-dimethylaminopyridine (DMAP, 98%, TCI America), 1-(3-Dimethylaminopropyl)-3-ethylcarbodiimide hydrochloride (EDC·HCl, 98%, Thermo Scientific), 6-(p-toluidinyl)naphthalene-2-sulfonic acid (TNS, Sigma-Aldrich), 1,2-dioleoyl-sn-glycero-3-phosphoethanolamine (DOPE, Avanti Polar Lipids), 1,2-distearoyl-sn-glycero-3-phosphocholine (DSPC, Avanti Polar Lipids), 1,2-dioleoyl-3-trimethylammonium-propane (DOTAP, Avanti Polar Lipids), D-Lin-MC3-DMA (MC3, MedChemExpress), cholesterol (Sigma-Aldrich) and 1,2-dimyristoyl-sn-glycero-3-phosphoethanolamine-N-[methoxy(polyethyleneglycol)-2000] (C14PEG2K, Avanti Polar Lipids) were used as received. Organic solvents were purchased from Fisher Scientific. Chloroform-d (CDCl$_3$) was purchased from Acros Organics.

### Biological reagents
Firefly luciferase (FLuc) mRNA (5moU) (L-7202), Cre mRNA (5moU) (L-7211) and Cas9 mRNA (5moU) (L-7206) were purchased from TriLink BioTechnologies. DNA barcode (b-DNA) design parameters followed our previous report[38]. A full list of b-DNA sequences can be found in Supplementary Data 2. Scrambled negative control sgRNA (Cat#A35526, Thermo Fisher Scientific) and VEGFR2 sgRNA (Synthego) were used as received. All oligonucleotides were purchased from Integrated DNA Technologies and were purified through standard desalting procedures. Luciferase 1000 Assay System (Ref. E4550) and

CellTiter-Glo Luminescent Cell Viability Assay Kit (Ref. G7572) were purchased from Promega Corporation. Anti-mouse CD31 antibody (AF488, Cat#102514, Clone#MEC13.3), CD45 antibody (Brilliant Violet 421, Cat#103134, Clone#30-F11), and EpCAM antibody (AF647, Cat#118212, Clone#G8.8) were purchased from BioLegend. Draq7 (Cat#424001, BioLegend) and 7AAD (Cat#A1310, Thermo Fisher Scientific) were used for live/dead staining. Tumor-bearing lung slices were stained with primary antibodies including goat anti-mouse/rat CD31 (Cat#AF3628, R&D Systems) and rabbit anti-mouse GFP (Cat#ab183734, abcam), followed by staining with secondary antibodies including AF488-conjugated donkey anti-rabbit (Cat#A-21206, Thermo Fisher Scientific) and AF555-conjugated donkey anti-goat (Cat#A32816, Thermo Fisher Scientific).

## Synthesis of aldehyde di-degradable tails

Taking 5-formyl-1,3-phenylene diheptanoate (A$_2$-6) as an example (Supplementary Fig. 1), briefly, 3,5-dihydroxybenzaldehyde (1.38 g, 10 mmol, 1.0 equiv), heptanoic acid (3.91 g, 30 mmol, 3.0 equiv), 1-(3-Dimethylaminopropyl)-3-ethylcarbodiimide hydrochloride (EDC·HCl, 5.75 g, 30 mmol, 3.0 equiv) and 4-dimethylaminopyridine (DMAP, 366 mg, 3 mmol, 0.3 equiv) were dissolved in anhydrous dichloromethane (DCM; 50 mL) and the mixture was cooled to 0 °C on an ice bath. The reaction was then allowed to warm to room temperature overnight. The supernatant was washed with HCl (1%; 2 × 50 mL), brine (2 × 50 mL), saturated sodium bicarbonate (2 × 50 mL) and brine (2 × 50 mL). The organic layer was collected, dried over Na$_2$SO$_4$, and concentrated in vacuo. The final pure monomer was further purified by flash chromatography (silica gel, hexane to DCM/hexane = 2/1) as light-yellow oil.

A$_2$-6 (Supplementary Fig. 2). $^1$H NMR (400 MHz, CDCl$_3$) $\delta$: 9.98 (s, 1H), 7.52 (s, 2H), 7.21 (s, 1H), 2.63–2.56 (m, 4H), 1.82–1.71 (m, 4H), 1.49–1.30 (m, 12H), 0.98–0.89 (m, 6H).

LC-MS (m/z): Calcd for [M + EtOH + Na]$^+$: 431.2, Found: 431.2 (Supplementary Data 3).

A$_2$-6b (Supplementary Fig. 3). $^1$H NMR (400 MHz, CDCl$_3$) $\delta$: 9.98 (s, 1H), 7.51 (s, 2H), 7.18 (s, 1H), 2.63–2.56 (m, 2H), 1.91–1.53 (m, 4H), 1.51–1.27 (m, 14H), 0.98–0.88 (m, 6H).

LC-MS (m/z): Calcd for [M + EtOH + Na]$^+$: 431.2, Found: 431.2 (Supplementary Data 4).

A$_2$-7 (Supplementary Fig. 4). $^1$H NMR (400 MHz, CDCl$_3$) $\delta$: 9.97 (s, 1H), 7.52 (s, 2H), 7.21 (s, 1H), 2.76–2.67 (m, 2H), 1.81–1.71 (m, 4H), 1.47–1.25 (m, 16H), 0.98–0.87 (m, 6H).

LC-MS (m/z): Calcd for [M + EtOH + Na]$^+$: 459.2, Found: 459.2 (Supplementary Data 5).

A$_2$-7b (Supplementary Fig. 5). $^1$H NMR (400 MHz, CDCl$_3$) $\delta$: 9.98 (s, 1H), 7.51 (s, 2H), 7.17 (s, 1H), 2.76–2.66 (m, 2H), 1.88–1.53 (m, 4H), 1.50–1.26 (m, 18H), 0.97–0.87 (m, 6H).

LC-MS (m/z): Calcd for [M + EtOH + Na]$^+$: 459.2, Found: 459.3 (Supplementary Data 6).

A$_2$-7b2 (Supplementary Fig. 6). $^1$H NMR (400 MHz, CDCl$_3$) $\delta$: 9.99 (s, 1H), 7.52 (s, 2H), 7.16 (s, 1H), 2.59–2.52 (m, 2H), 1.86–1.74 (m, 4H), 1.70–1.58 (m, 4H), 1.44–1.28 (m, 8H), 1.09–1.01 (m, 6H), 0.97–0.89 (m, 6H).

LC-MS (m/z): Calcd for [M + EtOH + Na]$^+$: 459.2, Found: 459.2 (Supplementary Data 7).

A$_2$-8 (Supplementary Fig. 7). $^1$H NMR (400 MHz, CDCl$_3$) $\delta$: 9.97 (s, 1H), 7.51 (s, 2H), 7.19 (s, 1H), 2.64–2.54 (m, 4H), 1.84–1.70 (m, 4H), 1.50–1.24 (m, 20H), 0.98–0.87 (m, 6H).

LC-MS (m/z): Calcd for [M + EtOH + Na]$^+$: 487.3, Found: 487.2 (Supplementary Data 8).

A$_2$-8b (Supplementary Fig. 8). $^1$H NMR (400 MHz, CDCl$_3$) $\delta$: 9.97 (s, 1H), 7.51 (s, 2H), 7.17 (s, 1H), 2.68–2.53 (m, 4H), 1.90–1.77 (m, 2H), 1.66–1.53 (m, 4H), 1.41–1.20 (m, 12H), 1.00–0.89 (m, 12H).

LC-MS (m/z): Calcd for [M + EtOH + Na]$^+$: 487.3, Found: 487.3 (Supplementary Data 9).

A$_2$-9 (Supplementary Fig. 9). $^1$H NMR (400 MHz, CDCl$_3$) $\delta$: 9.97 (s, 1H), 7.51 (s, 2H), 7.22 (s, 1H), 2.63–2.55 (m, 4H), 1.82–1.73 (m, 4H), 1.50–1.22 (m, 24H), 0.97–0.87 (m, 6H).

LC-MS (m/z): Calcd for [M + EtOH + Na]$^+$: 515.3, Found: 515.3 (Supplementary Data 10).

A$_2$-9b (Supplementary Fig. 10). $^1$H NMR (400 MHz, CDCl$_3$) $\delta$: 9.97 (s, 1H), 7.51 (s, 2H), 7.19 (s, 1H), 2.68–2.55 (m, 4H), 1.89–1.78 (m, 2H), 1.66–1.52 (m, 4H), 1.41–1.22 (m, 16H), 0.99–0.89 (m, 12H).

LC-MS (m/z): Calcd for [M + EtOH + Na]$^+$: 515.3, Found: 515.3 (Supplementary Data 11).

A$_2$-9b2 (Supplementary Fig. 11). $^1$H NMR (400 MHz, CDCl$_3$) $\delta$: 9.99 (s, 1H), 7.52 (s, 2H), 7.22 (s, 1H), 2.63–2.55 (m, 4H), 1.80–1.71 (m, 4H), 1.44–1.22 (m, 18H), 0.98–0.89 (m, 12H).

LC-MS (m/z): Calcd for [M + EtOH + Na]$^+$: 515.3, Found: 515.3 (Supplementary Data 12).

## Synthesis of aldehyde tri-degradable tails

Taking 2-formylbenzene-1,3,5-triyl tris(2-methylhexanoate) (A$_3$-6b) as an example (Supplementary Fig. 12), briefly, 2,4,6-trihydroxybenzaldehyde (1.54 g, 10 mmol, 1.0 equiv), heptanoic acid (5.86 g, 45 mmol, 4.5 equiv), 1-(3-Dimethylaminopropyl)-3-ethylcarbodiimide hydrochloride (EDC·HCl, 8.62 g, 45 mmol, 4.5 equiv) and 4-dimethylaminopyridine (DMAP, 550 mg, 4.5 mmol, 0.45 equiv) were dissolved in anhydrous dichloromethane (DCM; 80 mL) and the mixture was cooled to 0 °C on an ice bath, then the reaction was then allowed to warm to room temperature overnight. The supernatant was washed with HCl (1%; 2 × 50 mL), brine (2 × 50 mL), saturated sodium bicarbonate (2 × 50 mL) and brine (2 × 50 mL). The organic layer was collected, dried over Na$_2$SO$_4$, and concentrated in vacuo. The final pure monomer was further purified by flash chromatography (silica gel, hexane to ethyl acetate/hexane = 1/6) as orange oil.

A$_3$-6b (Supplementary Fig. 13). $^1$H NMR (400 MHz, CDCl$_3$) $\delta$: 10.15 (s, 1H), 6.93 (s, 2H), 2.84–2.69 (m, 3H), 1.96–1.76 (m, 6H), 1.61–1.19 (m, 21H), 0.98–0.83 (m, 9H).

LC-MS (m/z): Calcd for [M + Na]$^+$: 513.2, Found: 513.2.

A$_3$-7b (Supplementary Fig. 14). $^1$H NMR (400 MHz, CDCl$_3$) $\delta$: 10.15 (s, 1H), 6.90 (s, 2H), 2.83–2.69 (m, 3H), 1.91–1.71 (m, 6H), 1.47–1.22 (m, 27H), 0.98–0.88 (m, 9H).

LC-MS (m/z): Calcd for [M + Na]$^+$: 555.3, Found: 555.3.

A$_3$-7b2 (Supplementary Fig. 15). $^1$H NMR (400 MHz, CDCl$_3$) $\delta$: 10.15 (s, 1H), 6.91 (s, 2H), 2.66–2.56 (m, 3H), 1.91–1.77 (m, 6H), 1.73–1.59 (m, 6H), 1.46–1.36 (m, 12H), 1.10–0.89 (m, 18H).

LC-MS (m/z): Calcd for [M + Na]$^+$: 555.3, Found: 533.3.

A$_3$-8b (Supplementary Fig. 16). $^1$H NMR (400 MHz, CDCl$_3$) $\delta$: 10.06 (s, 1H), 6.95 (s, 2H), 2.73–2.53 (m, 3H), 1.92–1.74 (m, 6H), 1.66–1.47 (m, 6H), 1.44–1.15 (m, 21H), 1.02–0.85 (m, 18H).

LC-MS (m/z): Calcd for [M + Na + H]$^+$: 599.4, Found: 599.4.

A$_3$-9b2 (Supplementary Fig. 17). $^1$H NMR (400 MHz, CDCl$_3$) $\delta$: 10.15 (s, 1H), 6.66–6.61 (s, 2H), 2.69–2.34 (m, 6H), 1.82–1.69 (m, 6H), 1.68–1.60 (m, 3H), 1.43–1.21 (m, 24H), 0.99–0.86 (m, 18H).

LC-MS (m/z): Calcd for [M]$^+$: 616.8, Found: 616.8.

## Synthesis of cationic-degradable lipid libraries

A cationic degradable lipid library (180 lipids) was prepared by a one-pot, two step "Schiff base" reduction reaction between twelve different amine heads and fifteen different aldehyde degradable tails. Taking synthesis of 5-A$_2$-7b2 as an example, amine head **5** (17.63 mg, 0.2 mmol, 1 equiv) and A$_2$-7b2 (93.73 mg, 0.24 mmol, 1.2 equiv) were added in a glass vial equipped with a stir bar dissolved in ethanol. Then acetic acid (30 mg, 0.48 mmol, 2,4 equiv) was added into the above solution and the reaction was stirred at 80 °C for 3 h. Sodium borohydride (NaBH$_4$, 75 mg, 2 mmol, 10 equiv) was further added to react for 1 h at room temperature. Dichloromethane was added into the above solution, which was further washed by saturated sodium chloride (NaCl, x3) and dried by sodium sulfate (NaSO$_4$). The crude product was afforded by

removing the solvents and could then be used to screen the library for FLuc mRNA delivery without further purification.

Alternatively, the crude product could be further purified by flash chromatography (DCM to DCM/MeOH = 10/1).

5-A$_2$-7b2 (Supplementary Fig. 18). $^1$H NMR (400 MHz, CDCl3) $\delta$: 6.96 (s, 2H), 6.75 (s, 1H), 3.83 (s, 2H), 2.77–2.68 (m, 2H), 2.55–2.44 (m, 4H), 2.27–2.22 (s, 6H), 1.85–1.54 (m, 8H), 1.46–1.34 (m, 8H), 1.08–0.98 (m, 6H), 0.97–0.90 (m, 6H).

LC-MS (m/z): Calcd for [M + H]$^+$: 463.6, Found: 463.7 (Supplementary Data 13).

## Lipid nanoparticle formulation

All LNPs encapsulating mRNA used in this study were prepared as follows. An ethanol phase containing all lipids and an aqueous phase containing mRNA (FLuc mRNA, Cre mRNA, or Cas9 mRNA/sgRNA) were mixed using a microfluidic device to formulate LNPs. The ethanol phase contained CAD lipid, DOPE, cholesterol and C14-PEG2K, with a molar ratio of 35%, 16%, 46.5% and 2.5%. The aqueous phase was composed of mRNA dissolved in 10 mM citrate buffer. The weight ratio between Cas9 mRNA and sgRNA used was 4:1. The aqueous and ethanol phases were mixed at a flow rate of 1.8 mL/min and 0.6 mL/min (3:1) using a Pump33DS syringe pump (Harvard Apparatus, Holliston, MA). LNPs were then dialyzed in 1x PBS using a microdialysis cassette (20,000 MWCO, Thermo Fisher Scientific, Waltham, MA) for 2 h and filtered through a 0.45 μm filter. A Zetasizer Nano was used to measure the Z-average diameters, polydispersity index (PDI) and zeta potential. mRNA concentration and encapsulation efficiency of LNP formulations were measured using a modified Quant-iT RiboGreen (ThermoFisher) assay on a plate reader.

LNPs encapsulating DNA barcodes (b-DNA) and FLuc mRNA were prepared as follows. An ethanol phase contained CAD lipid, DOPE, cholesterol and C14-PEG2K, with a molar ratio of 35%, 16%, 46.5% and 2.5%. The aqueous phase was composed of b-DNAs and FLuc mRNA (w/w, 10:1) dissolved in 10 mM citrate buffer. LNPs were formulated by pipette mixing of the lipid solution into the nucleic acid-containing citrate buffer at a volume ratio of 1:3 (v/v). The resulting LNPs were dialyzed against 1x PBS in a 96-well microdialysis plate (10,000 MWCO, Thermo Fisher Scientific) at room temperature for 2 h. 50 μL of each LNP formulation was pooled together to make the LNP pool for dosing. DNA concentration in LNP formulations was determined by a NanoDrop Spectrophotometer. A Zetasizer Nano was used to measure the Z-average diameters, polydispersity index (PDI) and zeta potential.

Gold-standard lung-tropic MC3/DOTAP LNPs were used as a positive control following a similar formulation process, where the ethanol phase contained MC3 lipid (25%), 1,2-dioleoyl-3-trimethylammonium-propane (DOTAP, 50%), 1,2-distearoyl-sn-glycero-3-phosphocholine (DSPC, 5%), cholesterol (19.25%), and C14-PEG2K (0.75%).

## Characterization

$^1$H NMR spectrum were performed on a NEO 400 MHz spectrometer. LC-MS was performed on a Waters Acquity LCMS system equipped with UV-Vis and MS detectors. Flash chromatography was conducted on a Teledyne Isco CombiFlash Rf-200i chromatography system equipped with UV-Vis and evaporative light scattering detectors (ELSD). Particle size and zeta potential were measured by dynamic light scattering (DLS) with a Malvern Zetasizer Nano ZS. Particle morphology was measured by Cryo-TEM. Flow cytometry was performed using a FACSCanto or FACSymphony A3 instrument (BD Biosciences). In vitro luminescence intensity, pKa, encapsulation efficiency and mRNA concentration, and cell viability were quantified using an Infinite M Plex plate reader (Tecan, Morrisville, NC).

## Cell culture and animal studies

Dulbecco's Modified Eagle Medium (DMEM) was purchased from Gibco containing high glucose, L-glutamine, phenol red, and without sodium pyruvate and HEPES. Trypsin-EDTA (0.25%) and penicillin-streptomycin (P/S) were purchased from Gibco. Fetal bovine serum (FBS) was purchased from Sigma-Aldrich. HeLa cells (Cat#CCL-2, ATCC, Manassas, Virginia, USA) were cultured in DMEM supplemented with 10% FBS and 1% P/S. GFP expressing Lewis lung carcinoma (LLC-GFP) cells were provided by Ellen Puré Laboratory (University of Pennsylvania) and cultured in DMEM supplemented with 10% FBS and 1% P/S. LLC cells were obtained from ATCC (Cat#CRL-1642) and were transduced with GFP following a previous study[67].

All animal protocols were approved by the Institutional Animal Care & Use Committee (IACUC) of University of Pennsylvania (Protocol No. 806540), and were consistent with local, state and federal regulations as applicable. C57BL/6J (female, 6–8 weeks, 18–20 g) and B6.Cg-Gt(ROSA)26Sor$^{tm14(CAG-tdTomato)Hze}$/J (Ai14, female, 6–8 weeks, 18–20 g) mice were purchased from Jackson Laboratory. All mice were housed in a specific-pathogen-free animal facility at ambient temperature (22 ± 2 °C), air humidity 40–70% and 12 h dark/12 h light cycle and had free access to water and chow (Cat#5053, LabDiet). Animal health status was routinely checked by qualified veterinarians.

## In vitro FLuc mRNA LNP library screening

In a white transparent 96-well plate, HeLa cells were seeded at a density of $5 \times 10^3$ cells per well in 100 μL growth medium (DMEM, 10% FBS, 1% P/S) and incubated at 37 °C in 5% CO$_2$. The medium was exchanged for fresh growth medium, and then LNPs were added at a dose of 10 ng FLuc mRNA per well. Luciferase expression was measured 24 h after LNP transfection using a Luciferase Assay System (Promega) according to the manufacturer's protocol. The luminescence signal was normalized to PBS treated cells. Cell viability was measured using a CellTiter-Glo Luminescent Cell Viability Assay (Promega), in which the luminescence was normalized to PBS treated cells according to the manufacturer's protocol.

## In vivo barcoded LNP delivery

Mice were administered with a pool of different b-DNA LNPs, along with naked b-DNA (served as a negative control), at a dosage of 1.0 mg kg$^{-1}$ via tail vein injection. Tissues samples were harvested 6 h post-administration, snap-frozen in liquid nitrogen, disrupted into powder using a Geno/Grinder (SPEX SamplePrep) and stored at −80 °C until further analyzed.

## NGS library pool preparation

To prepare samples, approximately 30 μg of dry homogenized sample was suspended in a DNA-stabilizing lysis buffer containing 100 mM tris-HCl, 5 mM ethylenediaminetetraacetic acid (EDTA), 0.2% sodium dodecyl sulfate (SDS), and 200 mM NaCl. To remove protein and RNA contaminants, 20 μg of RNase A (New England Biolabs) and 100 μg of proteinase K (New England Biolabs) was added to each sample. Barcoded DNA (b-DNA) was subsequently extracted using a Zymo Oligo Clean and Concentrator kit (Zymo Research) following the manufacturer's instructions. Extracted b-DNA was amplified by PCR using Q5 High-Fidelity DNA Polymerase (New England Biolabs) with 16 denaturation-annealing-extension cycles using overhanging primers to add adapter (P5/P7) and index (i7) sequences for Illumina sequencing. The polymerase chain reaction (PCR) cleanup was performed using AMPure XP solid-phase reversible immobilization (SPRI) beads (Beckman Coulter Life Sciences) at a 1.8:1 bead:reaction volume ratio. Resultant library concentration was quantified using a Qubit 1X dsDNA High Sensitivity assay on a Qubit Flex fluorometer (Thermo Fisher Scientific). Libraries were combined in equimolar amounts to produce a library pool for next-generation sequencing (NGS), which was stored at −20 °C until sequencing. NGS was performed using an Illumina MiSeq series

sequencer (RRID:SCR_022382) with a 5% phiX sequencing control (Illumina) spike-in.

## NGS data analysis and visualization

NGS data were demultiplexed to produce FASTQ files using a standard Illumina sequencing workflow (bcl2fastq2). MD5 checksums were employed to ensure successful data transfer and data integrity. FASTQ files were processed using the UMI-tools Python package to extract unique molecular identifier (UMI) and barcode sequences[68]. All analysis downstream of sequence extraction used a combination of shell scripting and R scripts[69]. GNU sed and awk were used to extract barcode and UMI pairs to tabular data files. To collapse UMIs, an R script was employed using the Rncc, dplyr, multidplyr, stringi, and vroom packages[70–73]. Further data processing was performed using the dplyr, forcats, readxl, stringi, and tidyr packages[70,72,74–76]. Visualization was created using the ggplot2 package, with ggrepel used for labeling of enriched barcodes for hit identification[77,78]. The Nix package manager (with pinned nixpkgs revision 9cd622d1bfdced5b5b6425111d55ff20-de40649c) was used for dependency management for all analyses to maximize reproducibility, and GNU Make was used to orchestrate processing steps[79].

## Counterscreening of FLuc mRNA delivery in vivo

After analyzing in vivo b-DNA delivery, LNP-CAD3, LNP-CAD4, LNP-CAD9, LNP-CAD10, and LNP-CAD20 were selected for counterscreening by delivering FLuc mRNA. Mice were administered FLuc mRNA-LNP via tail vein injection. Luciferase expression was evaluated using an IVIS Spectrum imaging system (Caliper Life Sciences) 6 h post-injection. Mice were then injected with D-luciferin (PerkinElmer) at a dose of 150 mg/kg by intraperitoneal (i.p.) injection. Bioluminescence was quantified by measuring total flux in the region of interest where signal emanated using Living IMAGE Software provided by Caliper. Ex vivo imaging was performed on heart, liver, spleen, lung, and kidney after resection.

## Flow cytometry of tdTomato⁺ cell types in the lung

Ai14 mice were administered a single intravenous dose of Cre mRNA LNP-CAD9 at a dosage of 0.3 mg kg⁻¹ via tail vein injection. MC3/DOTAP LNPs formulated with Cre mRNA were used as a positive control. After 3 days post-injection, mice were first anesthetized by isoflurane, then perfused with 1x PBS. Afterwards the lung was collected, cut into small pieces, and digested by DMEM medium containing collagen IV (0.5 mg/mL). The above cell suspension was then filtered, centrifuged (5 min, 600 g) and lysed by ACK lysis buffer (1 mL) for 5 min. Single-cell suspensions were collected by centrifugation (5 min, 600 g) and resuspended in 1x PBS (500 µL), then stained by anti-mouse Alexa Fluor 488 CD31 antibody (1:200, BioLegend, Cat#102514), Brilliant Violet 421 CD45 antibody (1:200, BioLegend, Cat#103134), AF647 CD326 (EpCAM) antibody (1:200, BioLegend, Cat#118212) at 4 °C for 30 min. Lastly, the above suspensions were centrifuged and resuspended in Draq7 dyed 1x PBS (0.5 mL, 0.2%) for flow cytometry analysis.

## Immunofluorescence of the lungs

Mouse lung was obtained, transported to laboratory on ice and fixed with 3.2% PFA as previously reported[80]. Lung sections were then blocked in PBS + 1% BSA, 5% donkey serum, 0.1% Triton X-100, and 0.02% sodium azide for 1 h at room temperature, followed by incubation with primary antibody (CD31, BioLegend, Cat#102514) overnight at 4 °C. Afterwards, slides were washed and incubated with fluorophore-conjugated secondary antibody (Alexa Fluor 488-conjugated donkey anti-rat, 1:1000, Thermo Fisher Scientific, Cat#A-21208) for 2 h, which were further washed and incubated with DAPI for 5 min and mounted using ProLong Gold mountant (Life Sciences, Cat#P36930). Imaging was conducted with a Leica DMi8 microscope and analyzed with LAS X software (Leica).

## In vivo CRISPR-Cas9 VEGFR2 editing for antitumor therapy

LLC-GFP lung tumor model was established through tail veil injection of 1.0 × 10⁶ LLC-GFP cells into female C57BL/6J mice. On day 20 after tumor cell inoculation, mice were randomly assigned to four groups: PBS treated group (n = 12, G1), LNP-CAD9 encapsulating Cas9 mRNA/scramble sgRNA treated group (n = 12, G2), LNP-CAD9 encapsulating Cas9 mRNA/VEGFR2 sgRNA treated group (n = 12, G3), and MC3/DOTAP LNP encapsulating Cas9 mRNA/VEGFR2 sgRNA treated group (n = 12, G4). Mice were treated every other day for a total of 2 times (2.0 mg kg⁻¹ of RNA per injection). 6 of the mice in each group were euthanized 7 days after administration of LNPs, and their lungs were collected for analyses. The rest of the mice were subjected to survival evaluation. For survival analysis, mice were euthanized upon reaching a body weight loss exceeding 20% via carbon dioxide asphyxiation. The sequence of VEGFR2 sgRNA (5′-GTCCCGGTACGAGCACTTGT-3′) was used according to a previous study[60]. Tumor-bearing lung tissue were fixed with 3.2% PFA and then subjected for paraffin wax section. The obtained lung section slides were dewaxed, then blocked in PBS + 3% BSA, and 0.1% Triton X-10 for 30 mins at room temperature, followed by incubation with primary antibody against CD31 (Cat#AF3628, 1:200, R&D Systems) and GFP (Cat#ab183734, 1:200, Abcam) overnight at 4 °C. Afterwards, slides were washed and incubated with secondary antibody (AF488-conjugated donkey anti-rabbit, 1:1000, Thermo Fisher Scientific, Cat#A-21206; AF555-conjugated donkey anti-goat, 1:1000, Thermo Fisher Scientific, Cat#A32816) for 2 h, which were further washed and incubated with DAPI for 5 min and mounted. Imaging was conducted with a Leica DMi8 microscope and analyzed with LAS X software (Leica).

## Statistics and reproducibility

Two-sided Wilcoxon rank-sum test was used for the analysis of normalized accumulation of LNP formulations. A log-rank test was utilized for statistical analysis of Kaplan-Meier curves. One-way analysis of variance (ANOVA) followed by Dunnett's multiple comparison test was utilized for statistical analysis. All in vitro experiments were performed independently at least three times. In vivo barcoding experiments were performed with a cohort size of 5 female mice. IVIS imaging was performed with a cohort size of 3 female mice. Cre mRNA delivery was performed with a cohort size of 4 female mice. Tumor inoculation was performed using a cohort size of 12 female mice. All data are presented as mean ± s.e.m.

## Reporting summary

Further information on research design is available in the Nature Portfolio Reporting Summary linked to this article.

## Data availability

All relevant data supporting the findings of this study are available within the paper and Supplementary Information. Source data are provided with this paper.

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

## Acknowledgements

M.J.M. acknowledges support from a US National Institutes of Health (NIH) Director's New Innovator Award (DP2 TR002776), a Burroughs Wellcome Fund Career Award at the Scientific Interface (CASI), a US National Science Foundation CAREER Award (CBET-2145491), and the American Cancer Society (RSG-22-122-01-ET). A.E.V. acknowledges support from the National Institutes of Health (R01HL153539) and the Margaret Q. Landenberger Foundation. A.G.H., C.F.E., A.J.M., S.J.S. and R.E. are supported by NSF Graduate Research Fellowships (Award 1845298). Z. Xiao is supported by a CRI Irvington fellowship (Grant#: CRI4168) from Cancer Research Institute. The authors thank the Penn Genomic and Sequencing Core (RRID:SCR_022383) and The Wistar Institute for technical assistance. We further thank BioRender.com for providing a platform to create the schematics used in figures.

## Author contributions

L.X. and M.J.M. conceived the concept. L.X. designed and synthesized the CAD lipids used in this study. L.X., A.G.H., G.Z., Z.X, X.H., J.X., and R.E. performed the experiments. L.X., A.G.H., K.W., and M.J.M. wrote the manuscript. L.X., A.G.H., G.Z., Z.X., R.E., X.H., N.G., X.X., J.X., C.F.E., S.J.S., A.J.M., M.G.A., J.C., K.W., A.E.V., D.W., and M.J.M reviewed and edited the manuscript.

## Competing interests

L.X. and M.J.M. have filed a patent application on this research. D.W. is named on patents that describe the use of nucleoside-modified mRNA as a platform to deliver therapeutic proteins and vaccines. D.W. and M.G.A. are named on patents describing the use of lipid nanoparticles for nucleic acid delivery. The other authors declare no competing interests.
