## [Peer Review File · Nature Communications]

Reviewers' comments:

Reviewer #1 (Remarks to the Author):

Xue et al., report a barcoded screening system to identify cationic-degradable (CAD) lipid-like materials with high transfection efficiency in the desired cells in vitro and in vivo. Starting from 180 CAD lipids generated quickly by combinatorial chemistry the authors screened them for their transfection efficacy in vitro by delivering mRNA encoding firefly luciferase. Barcoding technology was then used to quantify a selected panel of CAD LNPs delivered DNA barcodes in vivo. This strategy allowed the identification of 21 CAD LNPs with lung-tropic efficacy. Notably, 4 LNPs had enriched accumulation in the lungs with depleted or not notably enriched delivery in the liver and spleen. The authors also characterize the top candidate, LNP-CAD9, which preferentially delivers mRNA to lung endothelial cells and test it in mice at the 0.3mg/Kg dose.

The manuscript is well written and clear. My main comments are the following:

- 1) b-DNA was included in 10:1 ratio with mRNA. What is the impact on LNP structure and mRNA delivery? Can the authors compare with mRNA only LNPs (eg mRNA encoded protein expression)?
- 2) CAD9 seems to result in lower expression in the lung compared to some of the other top LNPs. This may result in underestimation of expression in the liver
- 3) the authors highlight that LNP-CAD9 delivering Cre mRNA can preferentially edit lung endothelial cells at the clinically relevant dose. I am not sure that CRE is relevant to claim efficacy at a clinically relevant dose.
- 4) endothelial cells are in many parts of the body. Why expression would be limited to endothelial cells in the lung? Have the authors excluded editing in endothelial cells outside the lung?

Reviewer #2 (Remarks to the Author):

This is an overall solid paper that is well written, composed, and clear. There remains a need to identify and develop new lipid materials for extrahepatic delivery, and this paper introduces novel lipids with excellent properties for mRNA delivery across multiple outputs. I recommend to accept with minor revisions noted below.

1. Could the authors comment on the potential role of size in lung tropism, selectivity, or in potency? Since many LNPs were tested, and the Supporting Information reports all the DLS diameters already, the authors may be able to determine whether size affected delivery or not.
2. Figure 2b, which LNP is this? Same for Figure 2c. The caption states a "representative" LNP, but the authors might as well be specific to the exact CAD LNP formulation.
3. Figure 3d, is there any way to add identifying information to the heat map? Axis labels to somehow group the amine heads or aldehyde tails (apart from a random barcode number) would be informative to visualize potential trends (if possible to do).

4. I suggest to measure the pKa of relevant CAD LNPs. It would be interesting to know if the TNS LNP calculated pKa correlates with organ selectivity or not. I did not find reference to pKa in the paper. pKa could be a factor in delivery efficacy (endosomal escape) and/or tissue tropism.

Reviewers' comments:

Reviewer #1 (Report for the authors (Required)):

Xue et al., report a barcoded screening system to identify cationic-degradable (CAD) lipid-like materials with high transfection efficiency in the desired cells in vitro and in vivo. Starting from 180 CAD lipids generated quickly by combinatorial chemistry the authors screened them for their transfection efficacy in vitro by delivering mRNA encoding firefly luciferase. Barcoding technology was then used to quantify a selected panel of CAD LNPs delivered DNA barcodes in vivo. This strategy allowed the identification of 21 CAD LNPs with lung-tropic efficacy. Notably, 4 LNPs had enriched accumulation in the lungs with depleted or not notably enriched delivery in the liver and spleen. The authors also characterize the top candidate, LNP-CAD9, which preferentially delivers mRNA to lung endothelial cells and test it in mice at the 0.3mg/Kg dose. The manuscript is well written and clear.

Response: We are thankful for the reviewer's valuable time and positive comments on our paper. We have conducted additional experiments to address the reviewer's concerns.

My main comments are the following:

1. ~~b-DNA~~ was included in 10:1 ratio with mRNA. What is the impact on LNP structure and mRNA delivery? Can the authors compare with mRNA only LNPs (eg mRNA encoded

protein expression)?

Response: We thank the reviewer for the insightful suggestion. We employed 3-A₂-7b LNPs as a representative example to assess the potential impact of b-DNA on structural characteristics and mRNA delivery efficacy. The structural morphology of 3-A₂-7b LNPs encapsulating b-DNA/FLuc mRNA (with a weight ratio of 1:10) closely resemble that of 3A₂-7b LNPs encapsulating FLuc mRNA (Figure 2b and Supplementary Figure 19a). Notably, the average particle size was smaller when co-encapsulating b-DNA with mRNA, which is likely due to the presence of an LNP population encapsulating only the relatively small b-DNA (Figure 2c and Supplementary Figure 19b). Importantly, despite these variations in average particle size, there is no appreciable disparity in transfection efficacy between these two LNPs formulations *in vitro* (Supplementary Figure 19c). These parts have been added in our revised manuscript (Page 5, Line 35) as follows: “To investigate the potential influence of b-DNA on LNP structure and the efficacy of mRNA delivery *in vitro*, 3-A₂-7b LNP platform was used as a representative example to encapsulate b-DNA/FLuc mRNA (at a weight ratio of 10:1) and FLuc mRNA, respectively. These two LNPs exhibit similar structural morphology (Figure 2b and Supplementary Figure 19a), however, LNP carrying b-DNA/FLuc mRNA displayed a reduction in average particle size (Figure 2c and Supplementary Figure 19b), which is attributed to the smaller size of b-DNA compared to mRNA. Nonetheless, despite these differences in particle size, no appreciable distinction in transfection efficacy was observed between these two LNP formulations *in vitro* (Supplementary Figure 19c).”.

Supplementary Figure 19. (a) Structural morphology of 3-A₂-7b LNP encapsulating b-DNA/FLuc (with a weight ratio of 10:1) as visualized by Cryo-TEM. Scale bar: 100 nm. (b) Distribution of hydrodynamic diameter of 3-A₂-7b LNP encapsulating b-DNA/FLuc mRNA (with a weight ratio of 10:1) acquired by DLS. (c) Luciferase expression following treatment of HeLa cells with 3-A₂-7b LNP encapsulating FLuc mRNA and b-DNA/FLuc mRNA (10 ng luciferase mRNA, n = 5 replicates). Statistical significance in (c) was calculated using Student’s t test with unpaired design. $P > 0.05$, not significant (ns).

2. CAD9 seems to result in lower expression in the lung compared to some of the other top LNPs. This may result in underestimation of expression in the liver

Response: We thank the reviewer for providing insightful feedback. The enrichment analysis of barcodes in different organs indicated the distribution of barcoded nanoparticles, rather than the expression of cargoes (**Figure 3e**). While both LNP-CAD24 and LNP-CAD56 exhibited high enrichment of b-DNA accumulation within the lungs, it is noteworthy that LNP-CAD24 and LNP-CAD56 displayed substantial enrichment in the liver and spleen, respectively (**Figure 3e**). This observation indicates that these formulations may not be optimal choices for lung-targeted delivery due to their pronounced distribution in other organs. Consequently, we sought LNPs demonstrating a higher preference for lung distribution while showing either depleted or not notably enriched accumulation in the liver and spleen. Such formulations hold promise for lung-tropic LNP delivery. Based on this criterion, we identified LNP-CAD3, LNP-CAD4, LNP-CAD9, and LNP-CAD10 as primary candidates among the b-DNA screening for lung delivery. However, it is important to note that increased distribution of barcodes does not necessarily equate to higher expression levels of mRNA encoding for proteins. To further validate the mRNA delivery efficacy to the lungs, we formulated these top-ranking platforms to carry FLuc mRNA and conducted *in vivo* screening. By analyzing reporter gene luminescence in the lungs, liver, and spleen, LNP-CAD9 was identified as the leading candidate for lung-targeted mRNA delivery (**Figure 4b-4f**), exhibiting luciferase expression predominantly in the lungs (~90% of total luminescence flux). We have made this clarification in our revised manuscript (**Page 6**,

Line 38, Line 41) as follows “Lung-targeted delivery carriers demonstrated a pronounced preference for lung distribution, while their presence in the liver and spleen was either significantly reduced or not notably enriched.” and “Notably, LNP-CAD24 and LNP-CAD56 displayed substantial enrichment in the liver and spleen, which are not desired for lung-targeted delivery. Consequently, LNP-CAD3, LNP-CAD4, LNP-CAD9, and LNP-CAD10 emerged as primary candidates selected from the b-DNA screening for lung-specific delivery, as they demonstrated highly enriched accumulation in the lungs with depleted or not notably enriched delivery in the liver and spleen.”

3. the authors highlight that LNP-CAD9 delivering Cre mRNA can preferentially edit lung endothelial cells at the clinically relevant dose. I am not sure that CRE is relevant to claim efficacy at a clinically relevant dose.

Response: We thank the reviewer for the insightful feedback. Genetically engineered mouse models constitute valuable tools for the investigation of genetic disorders^{1,2}, among which, genetically engineered tdTomato reporter mice, such as Ai14 mice, have gained widespread utility in organ-specific gene editing applications. These mice feature a LoxP flanked stop cassette that effectively prevents expression of tdTomato protein³. LNPs delivering Cre mRNA into specific organs have the capability to excise the stop cassette, thereby enabling the activation of constitutive tdTomato expression, which allows for detection of gene edited cells⁴⁻⁶. As a surrogate for more intricate gene editing methodologies, LNPs delivering Cre mRNA in the lung of Ai14 mice offers a means to produce and evaluate genetic modifications within specific cell populations. Based on the lung endothelial cell transfection of LNP-CAD9, we conducted additional experiments and

investigated the *in vivo* CRISPR-Cas9 based gene editing of VEGFR2 in the lung endothelial cells of orthotopic lung tumor model for antiangiogenic cancer therapy. LNP-CAD9 co-delivering Cas9 mRNA and VEGFR2 sgRNA demonstrated remarkable *in vivo* antitumor therapy, as evidenced by a significant reduction in VEGFR2 levels, decreased tumor area per lung, prolonged survival, and a marked decrease in microvascular density within the tumor area, outperforming gold-standard MC3/DOTAP LNPs (**Figure 5**). We have added these discussions in our revised manuscript (**Page 7, Line 24; Page 10, Line**

12; Page 8, Line 7) as follows: “For this, we employed genetically engineered tdTomato reporter mice, an Ai14 (constitutive loxP-STOP-loxP-tdTomato) mouse model⁵¹, which have gained widespread utility in organ-specific gene editing applications^{52,53}. These mice feature a loxP flanked stop cassette that effectively prevents expression of tdTomato protein^{51,52,53}. LNPs delivering Cre mRNA into specific organs have the capability to delete the stop cassette, thereby producing tdTomato fluorescence only in transfected cells upon intracellular delivery of Cre recombinase mRNA^{28,40,54} (**Figure 4g**).” and “Genetically engineered tdTomato reporter mice, such as Ai14 mice, have gained widespread utility in organ-specific gene editing applications^{52,53}. LNPs delivering Cre mRNA into specific organs have the capability to excise the stop cassette, thereby enabling the activation of constitutive tdTomato expression, which allows for detection of gene edited cells^{28,40,54}. In this study, LNP-CAD9 delivering Cre mRNA were shown to efficiently edit lung endothelial cells at the clinically relevant dose of 0.3 mg kg⁻¹. Based on the endothelial cell tropism of LNP-CAD9, we further investigated therapeutic potential of this platform on orthotopic lung cancer model, through CRISPR-Cas9 gene editors-mediated *in vivo* gene editing of VEGFR2 in lung endothelial cells for antiangiogenic cancer therapy. LNP-CAD9 co-delivering Cas9 mRNA and VEGFR2 sgRNA demonstrated remarkable *in vivo* antitumor therapy, as evidenced by a significant reduction in VEGFR2 levels, decreased tumor area per lung, prolonged survival, and a marked decrease in microvascular density within the tumor area, outperforming gold-standard MC3/DOTAP LNPs.”

And” Exploring therapeutic potential of top performing LNPs for *in vivo* gene editing for antitumor therapy

As LNP-CAD9 demonstrated promising editing efficacy of lung endothelial cells *in vivo*, we assessed the therapeutic potential of this newly identified platform in vascular-relevant disease models. Angiogenesis is a complex and vital physiological process, playing roles in embryo development, wound healing, and collateral vessel formation^{55,56}. However, angiogenesis becomes aberrantly upregulated in tumorigenesis, supporting tumor progression by supplying essential oxygen and nutrients^{56,57}. Consequently, antiangiogenic therapy strategies aim to starve tumors by disrupting the delivery of these vital resources, thereby suppressing tumor growth, invasion, and metastasis^{56,57}. Among the numerous pro-angiogenic factors that have been discovered, vascular endothelial growth factor (VEGF) ranks among the most important⁵⁸. VEGF binds to its receptor VEGFR2, on tumor vascular endothelial cells, which subsequently promotes endothelial cell proliferation, migration and survival, leading to increased tumor vascularization through the growth of new blood vessels⁵⁸. Thus, the targeted disruption of VEGFR2 expression holds promise as an approach to inhibit the VEGF-VEGFR2 signaling pathway for

antiangiogenic cancer therapy.

We established a representative orthotropic lung cancer model by *i.v.* administering Lung Carcinoma cell lines expressing GFP (LLC-GFP) and evaluated *in vivo* antiangiogenic cancer therapy efficacy of LNP-CAD9 co-encapsulating Cas9 mRNA/VEGFR2 single guide RNA (sgRNA) (**Figure 5a**). MC3/DOTAP LNPs served as a benchmark lung-tropic LNP control. After tumor inoculation for 20 days, the mice were randomly allocated into four groups and received *i.v.* administration of PBS (G1), LNP-CAD9 co-delivering Cas9 mRNA/scrambled sgRNA (G2), LNP-CAD9 co-delivering Cas9 mRNA/VEGFR2 sgRNA (G3), or MC3/DOTAP co-delivering Cas9 mRNA/VEGFR2 sgRNA (G4), with a total RNA dosage of 4.0 mg kg^{-1} (administered in two days of 2.0 mg kg^{-1} each injection). Cas9/VEGFR2 sgRNA complex would induce double-strand breaks and insertions/deletions within the VEGFR2 locus, and thereby inhibiting the VEGF-VEGFR2 signaling pathway^{59,60} (**Figure 5b**). Seven days after the last administration, mice were euthanized, and their lungs were isolated to assess *in vivo* antitumor efficacy under the various treatment conditions. The mice that received LNPs co-delivering Cas9 mRNA/VEGFR2 sgRNA (G3 and G4) exhibited decreased VEGFR2 expression as quantified by RT-qPCR compared to PBS or scrambled sgRNA treated groups (G1 and G2) (**Figure 5c**). Furthermore, evaluations of the tumor area per lung and histological examinations *via* hematoxylin and eosin (H&E) staining revealed that the administration LNP-CAD9 co-delivering Cas9 mRNA/VEGFR2 sgRNA led to a significant reduction in lung tumor burden, outperforming MC3/DOTAP treated groups (**Figure 5d and 5e**). The survival analysis also demonstrated that administration of LNP-CAD9 co-delivering Cas9 mRNA/VEGFR2 sgRNA presented the highest tumor-inhibitory potential among all the treatments. This treatment regimen extended the median survival period from 32 days (G1) to 52 days (G3) (**Figure 5f**). Additionally, we assessed angiogenesis of tumor tissues by immunostaining vascular endothelial cells using CD31 antibody⁶¹ (**Figure 5g and 5h**). The results revealed a marked reduction in newly formed tumor blood vessels within the site after treatment with LNP-CAD9 co-delivering Cas9 mRNA/VEGFR2 sgRNA (**Figure 5g**). Quantification of microvascular density (MVD) in the tumor tissues showed that the administration of Cas9 mRNA/VEGFR2 sgRNA encapsulated by LNPs (G3 and G4) significantly reduced MVD compared to PBS or scrambled sgRNA treated groups (G1 and G2), indicating a pronounced antiangiogenic effect (**Figure 5h**). Importantly, mice treated with LNP-CAD9 co-delivering Cas9 mRNA/VEGFR2 sgRNA demonstrated superior antitumor efficacy compared to MC3/DOTAP LNPs. Collectively, these findings underscore the therapeutic potential of LNP-CAD9 platform for inhibiting tumor angiogenesis in the lung, resulting in effective suppression of tumor growth and far surpassing the lung-tropic gold-standard MC3/DOTAP formulations.”.

Figure 5. Antiangiogenic therapy through knocking out VEGFR2 by LNPs in orthotopic lung cancer model. (a) Schematic illustration of lung tumor implantation through *i.v.* injection of Lewis Lung Carcinoma cell lines expressing GFP (LLC-GFP) and treatment protocol of C57BL/6J mice. On day 20 after tumor cell inoculation, mice were randomly assigned to four groups: PBS treated control (G1), LNP-CAD9 encapsulating Cas9 mRNA/scrambled sgRNA treatment (G2), LNP-CAD9 encapsulating Cas9 mRNA/VEGFR2 sgRNA treatment (G3), and MC3/DOTAP LNP encapsulating Cas9

mRNA/VEGFR2 sgRNA treatment (G4). Mice were treated two days later for a total of 2 doses (2.0 mg kg⁻¹ of RNA per injection). The weight ratio of Cas9 mRNA and sgRNA was set as 4:1. Seven days after the last administration, 6 mice in each group were euthanized and their lungs were isolated for antiangiogenic analysis. The remaining mice were used for survival evaluation. (b) Antiangiogenic mechanism through LNP-mediated VEGFR2 knockout. LNPs encapsulating Cas9 mRNA/VEGFR2 sgRNA demonstrated the ability to reduce the expression of VEGFR2 in lung endothelial cells, which further inhibited the VEGF-VEGFR2 pathway. (c) RT-qPCR measurement of VEGFR2 level in different treatment groups. (3≤n≤4 mice) (d) Quantification of tumor areas per lung of different treatment groups. (n = 6 mice) (e) Representative H&E staining of lung tissue after sacrifice. Arrows indicate tumor areas in the lungs. Scale bar: 1 mm. (f) Percent survival of mice under different treatments. (n = 6 mice) (g) Representative immunostaining of tumor areas in the lungs. Endothelial cells were stained by CD31 antibody. DAPI was used for nuclear staining. (h) Microvascular density (MVD) in the tumor area under different treatments. (n = 6 mice). Statistical significance in (c), (d), (f), and (h) was calculated using one-way analysis of variance (ANOVA), followed by Dunnett's multiple comparison test. ****p* < 0.001; ***p* < 0.01; **p* < 0.05; *p* > 0.05, not significant. Data are presented as mean ± s.e.m.

4. endothelial cells are in many parts of the body. Why expression would be limited to endothelial cells in the lung? Have the authors excluded editing in endothelial cells outside the lung?

Response: We thank the reviewer for the invaluable feedback. We conducted additional experiments to evaluate if LNP-CAD9 can selectively edit endothelial cells in the lung, rather than in the liver and heart. Following administration of LNP-CAD9 encapsulating Cre mRNA into Ai14 mice model, different organs (lung, liver, and heart) were dissected and endothelial cell populations were isolated for flow cytometric analysis. Remarkably, our results demonstrated the absence of discernible editing of endothelial cells within the liver and heart, demonstrating the specific and targeted editing of lung endothelium achieved by LNP-CAD9 delivering Cre mRNA (**Supplementary Figures 24-25**). This has been integrated into our revised manuscript (**Page 7, Line 41**) as follows: "Moreover, LNP-CAD9

delivering Cre mRNA did not demonstrate discernible editing of endothelial cells within the liver and heart, highlighting the specific and targeted editing of lung endothelium achieved by LNP-CAD9 (**Supplementary Figures 24-25**)."

Supplementary Figure 24. (a) Representative gating strategy for tdTomato⁺ endothelial cells in the liver. 7AAD was used to distinguish live and dead cells. CD45 antibody was used to stain immune cells, then CD45⁻/CD31⁺ was used to identify liver endothelial cells. Ai14 mice was administered with PBS or LNP-CAD9 delivering Cre mRNA at a total dosage of 0.3 mg kg⁻¹. The mice were necropsied 3 days post-injection for flow cytometry studies. (b) Proportion of tdTomato⁺ endothelial cells in the liver assessed by flow cytometry. Data are presented as ± s.e.m. (n = 4 mice).

Supplementary Figure 25. (a) Representative gating strategy for tdTomato⁺ endothelial cells in the heart. 7AAD was used to distinguish live and dead cells. CD45 antibody was used to stain immune cells, then CD45⁻/CD31⁺ was used to identify heart endothelial cells. Ai14 mice was administered with PBS or LNP-CAD9 delivering Cre mRNA at a total dosage of 0.3 mg kg⁻¹. The mice were necropsied 3 days post-injection for flow cytometry studies. (b) Proportion of tdTomato⁺ endothelial cells in the heart assessed by flow cytometry. Data are presented as ± s.e.m. (n = 4 mice).

Reviewer #2 (Report for the authors (Required)):

This is an overall solid paper that is well written, composed, and clear. There remains a need to identify and develop new lipid materials for extrahepatic delivery, and this paper introduces novel lipids with excellent properties for mRNA delivery across multiple outputs. I recommend to accept with minor revisions noted below.

Response: We thank the reviewer for their time and overall positive feedback to our manuscript. We have conducted additional experiments and addressed all the comments below.

1. *Could the authors comment on the potential role of size in lung tropism, selectivity, or in potency? Since many LNPs were tested, and the Supporting Information reports all the DLS diameters already, the authors may be able to determine whether size affected delivery or not.*

Response: We thank the reviewer for the insightful comment. The size of nanoparticles is a pivotal determinant in achieving effective delivery to the tissues of interest. Depending on the delivery route, such as systemic administration and inhalation, nanoparticle size assumes varying importance. In the context of inhalation, it has been well-documented that particles with diameters exceeding 5 μm tend to deposit primarily in the upper airway due to impaction forces. In contrast, particles within the 1-5 μm range are notably efficient in reaching deep lung regions through inertial impaction and sedimentation mechanisms^{7,8}. Nanoparticles smaller than 1 μm can penetrate even further, reaching the alveoli through diffusion and sedimentation processes^{7,8}.

Conversely, when considering systemic administration, the relationship between LNP size and lung-tropic activity is relatively weak. Lung targeting is often achieved through other mechanisms, such as the formulation of protein corona on the LNP surface through endogenous interactions and active targeting strategies involving ligand attachment⁹. In our study, we conducted a detailed analysis of LNP size in relation to lung-tropic activity (**Supplementary Figure 21**). However, there is no discernible correlation between LNP size and their lung-tropic profile in our investigation. This suggests that other mechanisms, potentially involving endogenous targeting, may contribute significantly to the observed lung-tropic delivery of these barcoded LNPs. We have integrated this pertinent discussion in our revised manuscript (**Page 6, Line 32**) as follows: "We further investigated the correlation between LNPs size and their lung delivery efficacy, observing only a weak relationship between the LNP size and their lung-tropic activity (**Supplementary Figure 21**). The lung tropism of these LNPs may derive from alternative mechanisms such as endogenous targeting^{14,33}".

Supplementary Figure 21. Correlation of LNPs size and their respective lung delivery barcoding rank number.

2. Figure 2b, which LNP is this? Same for Figure 2c. The caption states a “representative” LNP, but the authors might as well be specific to the exact CAD LNP formulation.

Response: We thank the reviewer for their comment. We used 3-A₂-7b LNP for the experiments in **Figure 2b and 2c**. We add these details in the figure caption in our revised manuscript (**Page 15, Line 6, Line 7**) as follows: “(b) Representative cryogenic transmission electron microscopy (cryo-TEM) image of 3-A₂-7b LNP morphology. Scale bar: 100 nm. (c) Hydrodynamic size distribution of 3-A₂-7b LNP revealed by DLS.”.

3. Figure 3d, is there any way to add identifying information to the heat map? Axis labels to somehow group the amine heads or aldehyde tails (apart from a random barcode number) would be informative to visualize potential trends (if possible to do).

Response: We thank the reviewer for the insightful feedback. We have now updated **Figure 3d and Supplementary Figure 19** to include comprehensive information that enables the identification of each CAD lipid and its respective *in vivo* biodistribution from the heat map in our revised manuscript.

4. I suggest to measure the pKa of relevant CAD LNPs. It would be interesting to know if the TNS LNP calculated pKa correlates with organ selectivity or not. I did not find reference to pKa in the paper. pKa could be a factor in delivery efficacy (endosomal escape) and/or tissue tropism.

Response: We thank the reviewer for the insightful comments. We completely agree with the reviewer on this point. We conducted an assessment of the apparent pKa pertaining to representative liver-enriched LNPs (LNP-CAD20 and LNP-CAD95), lung-enriched LNPs (LNP-CAD10 and LNP-CAD3), and spleen-enriched LNPs (LNP-CAD14 and LNP-CAD73). Our investigations reveals that LNP-CAD20 and LNP-CAD95 have an apparent pKa that fall comfortably within the well-established range of 6 to 7 (**Supplementary Figure 27**), as is typical for liver-targeted LNPs^{10,11}. Conversely, LNP-CAD14 and LNP-CAD73 exhibit notably lower pKa between 4 to 5, a characteristic observed in spleen-targeted LNPs¹⁰ (**Supplementary Figure 27**). These findings are in concordance with earlier research concerning the relative pKa of LNPs with respect to tissue-specific activity¹⁰. However, in

the case of lung-enriched LNPs (LNP-CAD10 and LNP-CAD3), their pKa is measured as 4.905 and 5.806, respectively, which are not greater than the critical threshold of 9, a requirement for effective lung-targeting LNP delivery as previously reported¹⁰. Collectively, these results provide compelling evidence that the pKa of LNPs represents only one facet of the complicated landscape governing tissue-specific mRNA delivery activity. We have incorporated this discussion into our revised manuscript (**Page 10, Line 26**) as follows: “It has been reported that the apparent pKa of LNPs can exert influence over tissue-specific mRNA delivery activity. To investigate this, we selected representative liver-enriched (LNP-CAD20 and LNP-CAD95), lung-enriched (LNP-CAD10 and LNP-CAD3), and spleen-enriched (LNP-CAD14 and LNP-CAD73) LNPs for examination. In our investigation, CAD20 and LNP-CAD95 exhibited apparent pKa comfortably residing within the well-established range of 6 to 7 for liver-targeted LNPs, whereas LNP-CAD14 and LNP-CAD73 displayed lower pKa between ranging from 4 to 5 for spleen-targeted LNPs (**Supplementary Figure 27**). These findings align with prior research investigating the relative pKa of LNPs concerning tissue-specific activity^{33,62}. Nevertheless, in the case of lung-enriched LNPs (LNP-CAD10 and LNP-CAD3), our measurements revealed pKa of 4.905 and 5.806 (**Supplementary Figure 27**), respectively. Notably, these values do not exceed the pivotal threshold of 9, as stipulated for effective lung-targeting LNP delivery in previous reports³³. Cumulatively, these results provide compelling evidence that the pKa of LNPs represents only one facet of the complicated landscape governing tissue-specific mRNA delivery activity.”.

Supplementary Figure 27. Representative TNS assay curves for determining the apparent pKa of LNP-CAD20 (a), LNP-CAD95 (b), LNP-CAD10 (c), LNP-CAD3 (d), LNP-CAD14 (e) and LNP-CAD73 (f). Apparent pKa was defined as the point at which 50% of maximum TNS fluorescence was achieved.

References

1. Debster, J. D., Santagostino, S.F., Foreman, O. Applications and considerations for the use of genetically engineered mouse models in drug development. *Cell. Tissue. Res.*, **380**, 325 (2020).

2. Lee, H. Genetically engineered mouse models for drug development and preclinical trials. *Biomol. Ther.*, **22**, 267 (2014).
3. Staahl, B.T., et al. Efficient genome editing in the mouse brain by local delivery of engineered Cas9 ribonucleoprotein complexes. *Nat. Biotechnol.*, **35**, 431 (2017).
4. Cheng, Q., et al. Selective organ targeting (SORT) nanoparticles for tissue-specific mRNA delivery and CRISPR-Cas gene editing. *Nat. Nanotechnol.* **15**, 313 (2020).
5. Ni, H.Z., et al. Piperazine-derived lipid nanoparticles deliver mRNA to immune cells in vivo. *Nat. Commun.*, **13**, 4766 (2022).
6. Li, B.W., et al. Combinatorial design of nanoparticles for pulmonary mRNA delivery and genome editing. *Nat. Biotechnol.*, **41**, 1410 (2023).
7. Forest, V., Pourchez, J. Nono-delivery to the lung – by inhalation or other routes and why nono when micro is largely sufficient? *Adv. Drug. Deliv. Rev.*, **183**, 114173 (2022).
8. Tam, a., et al. Lipid nanoparticle formulations for optimal RNA-based topical delivery to murine airways. *Eur. J. Pharm. Sci.*, **176**, 106234 (2022).
9. Dilliard, S.A., Siegwart, D.J. Passive, active and endogenous organ-targeted lipid and polymer nanoparticles for delivery of genetic drugs. *Nat. Rev. Mater.* **8**, 282 (2023).
0. Dilliard, S.A., Cheng, Q., Siegwart, D.J. On the mechanism of tissue-specific mRNA delivery by selective organ targeting nanoparticles. *Proc. Natl. Acad. Sci. U. S. A.* **118**, e2109256118 (2021).
1. Akinc, A. et al. The Onpattro story and the clinical translation of nanomedicines containing nucleic acid-based drugs. *Nat. Nanotechnol.* **14**, 1084 (2019).

REVIEWERS' COMMENTS

Reviewer #2 (Remarks to the Author):

My comments have all been adequately addressed. I support this paper for acceptance and publication in Nature Communications in the current form.

Manuscript number: NCOMMS-23-22935A-Z

Authors' response to reviewers' comments

Reviewers' comments:

Reviewer #2 (Report to the Author):

My comments have all been adequately addressed. I support this paper for acceptance and publication in Nature Communications in the current form.

Response: We thank the reviewer for this encouraging decision. We thank the reviewer again for their time and effort in helping us improve the manuscript.